# NOT ALL LLM-GENERATED DATA ARE EQUAL: RETHINKING DATA WEIGHTING IN TEXT CLASSIFICATION

**Hsun-Yu Kuo**[*,1,2,3,†]**, Yin-Hsiang Liao**[*,1,2,†]**, Yu-Chieh Chao**[1]**, Wei-Yun Ma**[1,†,‡]**, Pu-Jen Cheng**[2,†]
[1]Academia Sinica [2]National Taiwan University
[3]Swiss Federal Institute of Technology in Lausanne (EPFL)
`hsun-yu.kuo@epfl.ch`,
`{zenonliao, vpj870331, ma}@iis.sinica.edu.tw`,
`pjcheng@csie.ntu.edu.tw`

## ABSTRACT

Synthetic data augmentation via Large Language Models (LLMs) allows researchers to leverage additional training data, thus enhancing the performance of downstream tasks, especially when real-world data is scarce. However, the generated data can deviate from the real-world data, and this misalignment can bring about deficient results while applying the trained model to applications. Therefore, we proposed efficient weighted-loss approaches to align synthetic data with real-world distribution by emphasizing high-quality and diversified data generated by LLMs using merely a tiny amount of real-world data. We empirically assessed the effectiveness of our methods on multiple text classification tasks, and the results showed that leveraging our approaches on a BERT-level model robustly outperformed standard cross-entropy and other data weighting approaches, providing potential solutions to effectively leveraging synthetic data from any suitable data generator.

## 1 INTRODUCTION

The quantity and quality of data play a significant role in many tasks of Natural Language Processing (NLP). However, due to the scarcity of data in a particular domain for a specific task, we may need expertise to collect such data, resulting in budget limitations. Fortunately, Large Language Models (LLMs) provide a practical solution to this problem. LLMs, such as GPT series (Brown et al., 2020; OpenAI, 2022; OpenAI et al., 2024), can be leveraged to generate synthetic data that mimics real-world examples, thereby enriching the training set (Wang et al., 2023). Taori et al. (2023), and other works (Ye et al., 2022; West et al., 2022; Li et al., 2023) have shown the capability of using LLM-generated data for the downstream tasks, and it seems to be a new cut-in solution to any NLP downstream tasks. However, training models with LLM-generated data can lead to drawbacks such as model collapse (Shumailov et al., 2023; Dohmatob et al., 2024), tail phenomena, reinforcing LM biases (Wang et al., 2023). Moreover, based on our empirical study, the performance of models trained on synthetic data without proper processing can be lower than models trained on much smaller real-world data (Sec. 3.1), highlighting the uncertainty of using LLM-generated data.

Previous works took filtering strategy to get high quality or variant data (Dubey et al., 2024; MetaAI, 2024; Chiang et al., 2023; West et al., 2022; Meng et al., 2022; 2023). Still, this strategy was mostly human-crafted or needed efforts to train a judge model, and most importantly, filtering strategies abandoned the potential of the filtered data that may contribute to the final performance. In contrast, data weighting approaches leverage all the training data, including augmented and biased data, but prioritize data by giving nonuniform weights to the loss of each data point. For example, Focal-Loss (Lin et al., 2017) prioritized more diverse data; Hu et al. (2019) and SunGen (Gao et al., 2023)

---

[*]Joint first authorship; [†]Data Science Degree Program, National Taiwan University and Academia Sinica; Work done while Hsun-Yu Kuo and Yu-Chieh Chao were at Academia Sinica; [‡]Corresponding author; Code: https://github.com/Hsun-Yu/DIMP-Loss

optimized the weights of training samples so that the model performs best on a small real-world dataset. It is worth noting that while Hu et al. (2019) and SunGen steered the training toward higher performance, using these methods in a large-scale scenario seems infeasible because the weights are regarded as learnable parameters, the number of which increases as the training set grows.

Thus, inspired by the objective of Hu et al. (2019), we introduce two novel, efficient, automatic weighted-loss approaches: Importance Loss (IMP-Loss) and Dynamic Importance Loss (DIMP-Loss), which are designed to closely align the distribution of synthetic data with that of real-world data. Furthermore, both IMP-Loss and DIMP-Loss incorporate mechanisms as *quality-checkers* and *diversity-checkers*, assigning higher quality and more diverse data points with greater weight. In other words, these methods prioritize data points that are more *relevant* and more *informative* to target downstream tasks, thereby reducing the impact of less valuable data on the fine-tuning model.

To validate our approaches, we conduct comprehensive empirical studies focusing on various text classification tasks by comparing the performance of models trained with our novel weighted-loss objectives under different conditions: 1) models trained exclusively on LLM-generated datasets using few-shot prompts from a limited real-world dataset (Sec. 5.1); 2) models trained on large, real-world datasets (Sec. 5.2); and 3) models trained on noisy datasets (Sec. G). Our findings indicate that using a small real-world dataset to build the *quality checkers* and incorporating *diversity checkers* highly enhances performance, even surpasses the few-shot prediction accuracy of the tremendous data generator (Sec. 5.1). This demonstrates the efficacy of our methods in leveraging little real-world data to improve models trained on LLM-generated datasets. Notably, DIMP-Loss is efficient in terms of model size (Sec. 5.1), data requirements (Sec. 5.1), and computational resources (Sec. 4.4), making it a practical solution to enhance downstream performance.

## 2 PRELIMINARIES

### 2.1 CROSS ENTROPY LOSS ON REAL-WORLD DATASET

In supervised learning for text classification tasks, we consider a real-world dataset $D_P = \{(\mathbf{x}_i, y_i)\}_{i=1}^{M}$ comprising $M$ samples. Each pair $(\mathbf{x}_i, y_i)$ is drawn independently and identically distributed (i.i.d.) from the joint distribution $P(\mathcal{X}, \mathcal{Y})$, where $\mathbf{x}_i$ represents the input sample and $y_i \in \mathcal{Y} = \{1, 2, \ldots, C\}$ is the corresponding class label. This setup forms the basis for training models using the empirical cross-entropy loss (CE-Loss), a standard loss function in such tasks. The CE-Loss over the entire dataset $D_P$ is calculated as follows:

$$\mathcal{L}_{\text{CE}}(\theta, D_P) = -\frac{1}{M} \sum_{i=1}^{M} \log \hat{P}(y_i|\mathbf{x}_i; \theta) \xrightarrow{p} \mathbb{E}_P \left[ -\log \hat{P}(y|\mathbf{x}; \theta) \right] \tag{1}$$

where $\hat{P}(y_i|\mathbf{x}_i; \theta)$ is the predicted probability of the model with parameters $\theta$ for the true class label $y_i$ given input $\mathbf{x}_i$. The CE-Loss converges in probability to the expected version of conditional cross-entropy under the true joint distribution $P(\mathcal{X}, \mathcal{Y})$ by the law of large numbers. This convergence is crucial because the minimizer of the CE-Loss occurs if and only if the predicted distribution $\hat{P}(y|\mathbf{x}; \theta)$ matches the true distribution $P(y|\mathbf{x})$.

### 2.2 WEIGHTED CROSS ENTROPY LOSS (WCE-LOSS)

WCE-Loss is a modification of the standard CE-Loss that assigns different weights to each data point. It is defined as:

$$\mathcal{L}_{\text{WCE}}(\theta, D_P, w) = -\frac{1}{N} \sum_{i=1}^{N} w_i \log \hat{P}(y_i|\mathbf{x}_i; \theta) \tag{2}$$

Here, $w_i$ represents the weight assigned to the $i$-th data point $(\mathbf{x}_i, y_i)$. A higher weight $w_i$ assigned to a data point $(\mathbf{x}_i, y_i)$ indicates the data point has a greater influence on the model's learning or adjustment of parameters, thereby being considered more important for the training process.

There have been several variants of the weight function, such as Focal Loss $\mathcal{L}_{\text{FL}}$ (Lin et al., 2017). It addressed class imbalance and reduced the impact of easily classified examples as its weight function

was defined as $w_i = (1 - \hat{P}(y_i|\mathbf{x}_i; \theta))^\gamma$, where $\gamma \geq 1$ was a focusing parameter that adjusted the rate at which easy examples were down-weighted. Research has shown that models trained with Focal Loss were better calibrated under the i.i.d. assumption and performed well under distribution shifts (Mukhoti et al., 2020). This made Focal Loss a promising baseline for evaluating our proposed weight function in the context of LLM-generated synthetic data training.

Additionally, a series of meta-learning approaches addressed these challenges by leveraging bi-level optimization to dynamically adjust weights based on each instance's contribution to the meta-set from the real world. These methods handled class imbalance, noisy labels, and augmented data by reweighting these instances based on their gradient direction or model outputs, providing a flexible mechanism for weighting data points (Ren et al., 2018; Hu et al., 2019; Gao et al., 2023). While effective, meta-learning-based approaches were computationally expensive, making them difficult to scale up to larger datasets or complex models. In contrast, our methods share the same objective of optimizing performance on real-world data but achieve it without meta-learning. This makes it more computationally efficient and scalable while still maintaining high performance.

## 3 OPTIMIZATION ON LLM-GENERATED DATASET

LLMs are capable of generating synthetic datasets (Lu et al., 2023; West et al., 2022; Li et al., 2023), denoted as $D_Q = \{(\mathbf{x}_i, y_i)\}_{i=1}^N$, sourced from the distribution $Q(\mathcal{X}, \mathcal{Y})$. This distribution is shaped by specific prompts comprising instruction prompts, system prompts, and few-shot examples that guide the LLM's output. This method offers a valuable alternative for acquiring training data, especially when access to real-world data is limited. Moreover, the relevance of $Q$ can be further refined by incorporating few-shot examples from a small real-world dataset $D_{P'}$, enhancing the utility and applicability of the synthetic data (Li et al., 2023).

The CE-Loss on the LLM-generated dataset converges to the expected cross-entropy under $Q$:

$$\mathcal{L}_{\text{CE}}(\theta, D_Q) \xrightarrow{p} \mathbb{E}_Q \left[ -\log \hat{P}(y|\mathbf{x}; \theta) \right] \tag{3}$$

A significant distributional shift between $Q$ and $P$ may lead to suboptimal predictive performance on real-world data.

### 3.1 UNCERTAINTY OF LLM-GENERATED DATA PERFORMANCE

Our empirical study, shown in Table 1, demonstrates notable variability in the performance of CE-Loss on LLM-generated datasets. Specifically, on the Financial (Malo et al., 2014) and MRPC (Wang et al., 2018) benchmarks, CE-Loss on large LLM-generated datasets ($> 3$k samples) performs worse than training on small real-world datasets, which contain only around 200-400 samples. In contrast, CE-Loss in LLM-generated data improves accuracy for the Twitter Irony (Van Hee et al., 2018) benchmark. This variability underscores the uncertainty associated with using CE-Loss on LLM-generated data. These findings are consistent with another research (West et al., 2022), showing that when using CE-Loss, without proper filtering, LLM-generated data may lead to decent results on downstream tasks, even though its size is considerably larger than that of real-world data.

### 3.2 POTENTIAL OF LLM-GENERATED DATA: MODEL-BASED INFORMATION MEASUREMENT

We employ information-theoretic metrics to evaluate the uncertainty within the conditional distributions of real-world data and LLM-generated data. A higher **conditional entropy** indicates a more significant uncertainty given an input $\mathbf{x}$, suggesting various outcomes. We estimate this by separately fine-tuning a BERT model (Devlin et al., 2019) on both datasets. Higher conditional entropy is often associated with greater diversity within the dataset, reflecting a broader range of information that the model must learn to predict accurately. The **conditional KL divergence**[1] quantifies the difference between two conditional distributions, $P(y|\mathbf{x})$ and $Q(y|\mathbf{x})$, showing how well a model trained on one dataset describes another.

---

[1]It is also called conditional divergence or conditional relative entropy.

We show these metrics for a financial benchmark scenario. The real-world dataset $D_P$ exhibits significantly lower **conditional entropy** ($\mathbb{H}_P(y|\mathbf{x}) = 0.0365$) compared with the LLM-generated dataset $Q$ ($\mathbb{H}_Q(y|\mathbf{x}) = 0.2299$), indicating that $D_Q$ is more *diverse*. Furthermore, the **conditional KL divergence** from $P$ to $Q$ ($D_{\mathrm{KL}}(Q||P) = 1.8781$) is much greater than it from $Q$ to $P$ ($D_{\mathrm{KL}}(P||Q) = 0.444$), suggesting that models trained on real-world data struggle to capture the complexity of the synthetic dataset. Those models trained on the synthetic dataset are relatively efficient, requiring fewer additional nits on average to encode samples from the real-world distribution $P$. This difference, along with the results from Sec. 3.1 highlights that, although the synthetic dataset contains some points that are less representative of the real-world distribution, it still includes a substantial proportion of *relevant data points.* This analysis indicates the potential to improve modeling techniques to utilize LLM-generated data's rich, informative content.

### 3.3 PROBLEM FORMULATION

In this study, we devise a weighted loss function that transforms CE-Loss from an LLM-generated data distribution $Q$ to match the real-world data distribution $P$. We assumed the dataset $D_Q$ is i.i.d. and the LLM can approximate the real-world input distribution $P(\mathbf{x})$ through strategic prompting, effectively simulating $Q(\mathbf{x}) \approx P(\mathbf{x})$. For example, by using system prompts like *"Now you are a journalist writing news articles,"* it can produce synthetic texts that closely mimic authentic news articles. Lastly, we use a small set $D_{P'}$, approximately 200-400 samples from real-world datasets, to facilitate the alignment process. These samples are i.i.d. from the distribution $P$. We use $P'$ as the probability function representing this small real-world dataset.

This approach leverages the rich diversity of LLM-generated data to bridge the distributional gap between $Q$ and $P$. By creating an effective weighted loss function, we aim to enhance model performance on real-world tasks by better aligning synthetic data with real-world distributions.

## 4 METHODOLOGIES

In this section, we present our Importance Loss (IMP-Loss) and Dynamic Importance Loss (DIMP-Loss) methods, which transform the CE-Loss to align with the real-world distribution $P$ from the LLM-generated distribution $Q$.

### 4.1 IMP-LOSS: TRANSFORMATION FROM $Q$ TO $P$

To achieve convergence to the real-world data distribution $P$, we applied WCE-loss. Inspired by the Monte Carlo method of Importance Sampling (Hesterberg, 1995), used to estimate expectation values from a source distribution to a target distribution, we design the weight function as follows:

$$w_i = \frac{P(y|\mathbf{x}_i)}{Q(y|\mathbf{x}_i)} \tag{4}$$

By applying this weight function to WCE-Loss, the asymptotic convergence is approximately the expectation under $P$ (details in Appendix B):

$$
\begin{aligned}
\mathbb{E}_Q &\left[ -\frac{P(y|\mathbf{x})}{Q(y|\mathbf{x})} \log \hat{P}(y|\mathbf{x}; \theta) \right] \\
&= -\sum_{\mathbf{x} \in \mathcal{X}} Q(\mathbf{x}) \sum_{y \in \mathcal{Y}} P(y|\mathbf{x}) \log \hat{P}(y|\mathbf{x}; \theta) \\
&\approx -\sum_{\mathbf{x} \in \mathcal{X}} P(\mathbf{x}) \sum_{y \in \mathcal{Y}} P(y|\mathbf{x}) \log \hat{P}(y|\mathbf{x}; \theta) \\
&= \mathbb{E}_P \left[ -\log \hat{P}(y|\mathbf{x}; \theta) \right]
\end{aligned}
\tag{5}
$$

The approximation in the penultimate step is based on the assumption stated in Sec. 3.3: the LLM can simulate the real-world input distribution through careful and appropriate prompting. This transformation ensures that the WCE-Loss effectively aligns $Q$ with $P$.

Further, $Q$ can be estimated by fitting a neural model $\hat{Q}$, such as BERT, on the LLM-generated dataset $D_Q$ using the CE-Loss; however, estimating the weight function is challenging because the

real-world distribution $P$ is unknown. To address this, we fit a model $\hat{P}'$ on small real-world dataset $D_{P'}$. Using $\hat{P}'$ and $\hat{Q}$, we define the **Importance Loss** $\mathcal{L}_{\text{IMP}}(\theta, D_Q)$ as follows:

$$\mathcal{L}_{\text{IMP}}(\theta, D_Q) = -\frac{1}{N} \sum_{i=1}^{N} \frac{\overbrace{\hat{P}'(y_i|\mathbf{x}_i)}^{\text{Quality Checker}}}{\underbrace{\hat{Q}(y_i|\mathbf{x}_i)}_{\text{Diversity Checker}}} \log \hat{P}(y_i|\mathbf{x}_i; \theta) \tag{6}$$

Algorithm 1 outlines how we use IMP-Loss.

---

**Algorithm 1** Training with Importance Loss

---

**Require:** Small real-world dataset $D_{P'}$, synthetic dataset $D_Q$, model $\hat{P}$, initial parameters $\theta$

    **Step 1:** $\hat{P}' \leftarrow$ Estimation of $P'$ by fitting a model with CE-Loss on $D_{P'}$

    **Step 2:** $\hat{Q} \leftarrow$ Estimation of $Q$ by fitting a model with CE-Loss on $D_Q$

    **Step 3:** Compute the weights $w_i = \frac{\hat{P}'(y|\mathbf{x})}{\hat{Q}(y|\mathbf{x})}$ for each training sample $(\mathbf{x}, y) \in D_Q$

    **Step 4:** Optimize model parameters $\theta$ to minimize $\mathcal{L}_{\text{IMP}}(\theta, D_Q)$ by SGD

---

### 4.2 DIMP-Loss: Which data point causes the model to be closest to $P$?

In this section, drawing inspiration from online batch selection methods (Deng et al., 2023; Mindermann et al., 2022), we investigate *which data point in $D_Q$, when used for training, will most effectively bring the distribution of the model closer to $P$ in the subsequent optimization step*. In optimization formulation, this can be expressed as:

$$(\mathbf{x}^*, y^*) = \underset{(\mathbf{x}', y') \in D_Q}{\arg\min} \ \mathbb{E}_P \left[ -\log \hat{P}\left(y|\mathbf{x}; \theta_t, \{(\mathbf{x}', y')\}\right) \right], \tag{7}$$

where $\theta_t$ represents the model parameters at optimization step $t$. Consider a one-step optimization algorithm $f$ (e.g. SGD), where $\theta_{t+1} \leftarrow f(\theta_t, \{(\mathbf{x}', y')\})$. The algorithm updates the model parameters $\theta_t$ using $(\mathbf{x}', y')$ to obtain the new parameters $\theta_{t+1}$ after one optimization step. The Eq. 7 means the data point $(\mathbf{x}^*, y^*)$ is the optimal data point in $D_Q$ that leads to the lowest conditional cross-entropy after one update step. Specifically, it identifies which data point is used for training results in the model parameters that yield the model closest to the real-world distribution $P$.

In empirical settings, we may not have access to the complete real-world distribution $P$, but we can approximate it by a small real-world dataset $D_{P'}$, also denoted as $(\mathbf{y}_{P'}, \mathbf{X}_{P'})$ in the perspective of labels and inputs. This allows us to rewrite the objective as maximizing the probability:

$$\underset{(\mathbf{x}, y) \in D_Q}{\arg\max} \hat{P}(\mathbf{y}_{P'}|\mathbf{X}_{P'}; \theta_t, \{(\mathbf{x}, y)\}) = \underset{(\mathbf{x}, y) \in D_Q}{\arg\max} \frac{\hat{P}(y|\mathbf{x}; \theta_t, D_{P'})}{\hat{P}(y|\mathbf{x}; \theta_t)} \tag{8}$$

Eq. 8 aims to maximize the joint likelihood of all data points in $D_{P'}$. The joint likelihood involves inferring all data points in $D_{P'}$ and multiplying their prediction probabilities (due to the i.i.d. assumption). However, this optimization is infeasible, as it requires updating the model for each data point in $D_Q$, resulting in $|D_Q|$ models, and each needs evaluation on the whole $D_{P'}$.

Notably, by applying Bayes' rule, we derive the right-hand side of Eq. 8 (see Appendix C for details), showing a more feasible calculation approach. This requires evaluating only two models for each data point in $D_Q$: the denominator $\hat{P}(y|\mathbf{x}; \theta_t)$ is the current model in step $t$, and the numerator $\hat{P}(y|\mathbf{x}; \theta_t, D_{P'})$ would require additional training on $D_{P'}$. To simplify, we approximate $\hat{P}(y|\mathbf{x}; \theta_t, D_{P'})$ with $\hat{P}'(y|\mathbf{x})$, the probability estimated from $D_{P'}$ as in Deng et al. (2023).

The approximation of Eq. 8 is then utilized as the weight in our loss function. Consequently, if a data point brings the model closer to the real-world distribution $P$, its corresponding weight will be higher, thus having a greater impact on training. Thus, we define the **Dynamic Importance Loss**

**(DIMP-Loss)** $\mathcal{L}_{\text{DIMP}}(\theta_t, D_Q)$ as:

$$\mathcal{L}_{\text{DIMP}}(\theta_t, D_Q) = -\frac{1}{N} \sum_{i=1}^{N} \frac{\overbrace{\hat{P}'(y_i|\mathbf{x}_i)}^{\text{Quality Checker}}}{\underbrace{\hat{P}(y_i|\mathbf{x}_i;\theta_t)}_{\text{Diversity Checker}}} \log \hat{P}(y_i|\mathbf{x}_i;\theta_t) \tag{9}$$

The approximation of Eq. 8 simplifies the calculation of the weight function, making the implementation of DIMP-Loss practical. We can observe this weight function dynamically changes at each optimization step and adjust the weights based on the current parameters $\theta_t$, thereby continually refining the alignment between the model $\hat{P}_\theta$ and the real-world data distribution $P$. Algorithm 2 outlines how DIMP-Loss is used in training a model.

---

**Algorithm 2** Training with DIMP-Loss

---

**Require:** Small real-world dataset $D_{P'}$, synthetic dataset $D_Q$, model $\hat{P}$, initial parameters $\theta$
   **Step 1:** $\hat{P}' \leftarrow$ Estimation $P'(y|\mathbf{x})$ by fitting a model with CE-Loss on $D_{P'}$
   **Step 2:** Compute the $\hat{P}'(y|\mathbf{x})$ for each training sample $(\mathbf{x}, y) \in D_Q$
   **Step 3:** Optimize model parameters $\theta$ to minimize $\mathcal{L}_{\text{DIMP}}(\theta, D_Q)$ by SGD

---

To better understand the properties of DIMP-Loss, we derived a lower bound for it (details can be found in Appendix D). Precisely, we have:

$$\mathcal{L}_{\text{DIMP}}(\theta_t, D_Q) \geq \underbrace{-\frac{2}{N} \sum_{i=1}^{N} \hat{P}'(y_i|\mathbf{x}_i) \log \hat{P}(y_i|\mathbf{x}_i;\theta_t)}_{\text{Empirical distilled cross-entropy loss}} + \underbrace{\frac{1}{N} \sum_{i=1}^{N} \hat{P}(y_i|\mathbf{x}_i;\theta_t) \log \hat{P}(y_i|\mathbf{x}_i;\theta_t)}_{\text{Maximum entropy regularizer}} \tag{10}$$

DIMP-Loss can be interpreted as an upper bound on the regularized empirical distilled risk (Menon et al., 2021; Wang et al., 2022), where the "teacher" model is the quality checker. The regularizer is a maximum entropy term designed to prevent overconfidence in output distribution (Pereyra et al., 2017). In this context, DIMP-Loss can also be viewed as a form of knowledge distillation, where the knowledge from a model is trained on a small amount of real-world data. The objective is to align the predicted distribution $\hat{P}_\theta$ with $P'$ while promoting higher entropy in $\hat{P}_\theta$ to avoid overly confident predictions.

### 4.3 QUALITY AND DIVERSITY CHECKERS IN IMP-LOSS AND DIMP-LOSS

According to both Eq. 6 and 9, a high weight means the data point significantly influences the model. The **Quality Checker** ($\hat{P}'(y_i|\mathbf{x}_i)$) assesses the likelihood of a data point sampled from the real-world distribution $P$. Higher values indicate the data point is, highly relevant, and unambiguous for the real-world distribution $P$.

The **Diversity Checker** differs in the two losses, $\hat{Q}(y_i|\mathbf{x}_i)$ for IMP-Loss, and $\hat{P}(y_i|\mathbf{x}_i;\theta_t)$ for DIMP-Loss. In the context of IMP-Loss, a low **Diversity Checker** value $\hat{Q}(y_i|\mathbf{x}_i)$ suggests the data point contains a high amount of information within the LLM-generated dataset $D_Q$, because a redundant data point in $D_Q$ will have a high probability, indicating less diversity. Hence, it serves as an indicator of diversity from the perspective of the LLM-generated distribution. In contrast, for DIMP-Loss, a low **Diversity Checker** value $\hat{P}(y_i|\mathbf{x}_i;\theta_t)$ implies the data point is challenging to be learned in previous steps, departing from the data points the model has already learned. Thus, **Diversity Checker** of DIMP-Loss reflects diversity from the perspective of a model. This distinction highlights how each loss function prioritizes different aspects of data diversity during training. We simulated defect and redundant situations for further exploration in the Appendix. G.3.

### 4.4 COMPUTATIONAL COST OF TRAINING WITH IMP-LOSS AND DIMP-LOSS

The analysis covers computational requirements and practical run-time detailed in Appendix E.

**IMP-Loss.** According to Algorithm 1, the computational cost of training with IMP-Loss is approximately (ignore the cost on $D_{P'}$) twice training plus twice forward pass on $D_Q$. First, we estimate $P'(y|\mathbf{x})$ by fitting a model on $D_{P'}$, and $Q(y|\mathbf{x})$ by fitting a model on $D_Q$, respectively. Second, we compute the weights for each sample in $D_Q$ by $\hat{Q}(y|\mathbf{x})$ and $\hat{P}'(y|\mathbf{x})$. Although the estimation of $P'(y|\mathbf{x})$ incurs minimal cost due to the small size of $D_{P'}$, the primary additional overhead comes from the repeated training on $D_Q$ and the additional forward passes needed to compute the weights.

**DIMP-Loss.** According to Algorithm 2, the computational cost of training with DIMP-Loss is approximately (ignore the cost on $D_{P'}$) one training plus one forward pass on $D_Q$. On the one hand, we need to fit a model on the small real-world $D_{P'}$ to estimate $P'(y|\mathbf{x})$, the numerator of the weight coefficient, for each data point in $D_Q$. On the other hand, we compute the weights for each sample in $D_Q$ using the DIMP-Loss formulation, which involves evaluating the computed $\log \hat{P}(y_i|\mathbf{x}_i; \theta_t)$ and hence getting $\hat{P}(y_i|\mathbf{x}_i; \theta_t)$. Without the estimation to $Q(y|\mathbf{x})$, DIMP-Loss is more efficient than IMP-Loss. The computational overhead is only slightly higher than that of CE-Loss, because of the additional step of estimating $P'$ from the small dataset $D_{P'}$ and performing a single inference pass on $D_Q$, as the values of the quality checker for each data point remain constant in training.

## 5 EXPERIMENTS

| Dataset | Method | Financial | | Tweet Irony | | MRPC | |
|---|---|---|---|---|---|---|---|
| | | Acc | F1 | Acc | F1 | Acc | F1 |
| | GPT-3.5 few-shot | 79.46 | 81.6 | 63.39 | 69.39 | 69.28 | 71.75 |
| Small real world | CE-Loss (quality checker) | 78.05 | 75.26 | 62.5 | 62.38 | 73.16 | 68.69 |
| | Focal-Loss | 78.47 | 76.2 | 67.73 | 62.32 | 73.10 | 66.64 |
| | DIMP-Loss (Ours) | **79.87** | **77.05** | **69.01** | **67.05** | **74.84** | **66.80** |
| GPT-3.5 generated | CE-Loss | 77.39 | 74.01 | 76.91 | 76.8 | 72 | 65.47 |
| | Focal-Loss | 79.29 | 75.32 | 74.87 | 74.82 | 72.17 | 62.77 |
| | Hu et al.'s | 71.7 | 61.93 | 71.42 | 70.18 | 67.13 | 50.08 |
| | SunGen | 80.45 | 76.87 | 78.96 | 75.06 | 71.65 | 66.08 |
| | IMP-Loss (Ours) | 82.09 | **79.40** | **81.89** | **81.71** | **75.83** | **70.52** |
| | DIMP-Loss (Ours) | **82.67** | **79.53** | 78.44 | 78.14 | **75.83** | **70.04** |
| | - w/o diversity checker | 81.35 | 77.94 | 77.68 | 77.62 | 74.72 | 69.34 |
| Large real world | CE-Loss | 84.74 | 82.69 | 68.75 | 68.41 | 80.92 | 77.73 |
| | Focal-Loss | **84.98** | 81.98 | 67.6 | 67.19 | 80.35 | 76.28 |
| | Hu et al.'s | 80.19 | 76.58 | 60.33 | 37.63 | 71.36 | 67.78 |
| | SunGen | 84.65 | 82.51 | 63.9 | 62.66 | 80.81 | 78.78 |
| | IMP-Loss (Ours) | **85.3** | **83.27** | **70.15** | **70.08** | 81.33 | 78.3 |
| | DIMP-Loss (Ours) | **85.4** | **82.79** | 69 | 68.78 | **82.84** | **80.49** |

Table 1: Performance metrics across datasets and methods. The table showcases each combination's accuracy (Acc) and macro F1 score (F1). The methods include GPT-3.5 few-shot, CE-Loss, Focal-Loss, Hu et al.'s method, SunGen, IMP-Loss, and DIMP-Loss. Bold entries denote the performance within 0.5%, compared to the best performance of each training source.

We assessed our proposed methods by comparing them with standard loss functions across several text classification benchmarks, including Financial Phrasebank (Financial) (Malo et al., 2014), irony detection (Tweet Irony) (Van Hee et al., 2018), and the MRPC dataset from GLUE (Wang et al., 2018). Detailed descriptions and specifications are provided in Appendix H. In our experiments, we referred the large real-world data $D_P$ to the original training set from each benchmark and the small real-world data $D_{P'}$ to the original development set, with the sizes from approximately 200 to 400, as shown in Table 5. Our experiments explored three different scenarios: training solely on synthetic data (Sec. 5.1), real-world data (Sec. 5.2), and noisy data (Sec. G). We evaluated Accuracy (Acc) and Macro F1 score (F1) for every benchmark. These metrics were computed by comparing the model's predictions with the gold labels provided in the test sets. We used a BERT-based model for fine-tuning and building the checkers. The Appendix F details the configurations.

**Baselines.** CE-Loss, Focal Loss, SunGen and Hu et al. (2019) are our baselines, detailed in Sec. 2.1 and Sec. 2.2. Focal Loss addressed class imbalance and mitigated easily classified

data's impact, preventing overconfidence. The weight function for Focal Loss was defined as $w_i = (1 - \hat{P}(y_i|\mathbf{x}_i; \theta))^\gamma$ where $\gamma$ controlled the downweighting of easy examples. Mukhoti et al. (2020) showed models trained with Focal Loss exhibited better calibration under the i.i.d. assumption and performed robustly under distribution shifts. This made Focal Loss a strong baseline for evaluating our proposed approaches. Both SunGen and Hu et al. (2019) are bilevel optimization methods but differ in their weight update mechanisms and the objective functions of their outer loops. They used meta-learning to dynamically adjust weights training data by maximizing the likelihood on a small real-world dataset, similar to IMP-Loss and DIMP-Loss; however, our methods directly adjust weights based on quality and diversity checkers, while Hu et al.'s method and SunGen relied on meta-learning to optimize weights indirectly [2].

## 5.1 TRAINING ON LLM-GENERATED DATA

We compared our proposed methods with standard loss functions on LLM-generated data.

**Data Generation.** We used GPT-3.5-turbo-1106 (OpenAI, 2022), given a system prompt, 8-shot examples from the development set $D_{P'}$, and the corresponding labels to generate the input text. For the Financial and Tweet Irony, our generation prompt based on previous research (Li et al., 2023). Similarly, for the MRPC benchmark, the prompt included pairs of sentences with answers, which automatically guided the LLM in generating the answers. See Appendix I.1 for details.

**IMP-Loss and DIMP-Loss Outperform on LLM-Generated Data.** As shown in Table 1, our methods outperformed all baselines in all benchmarks. For instance, in Financial Phrasebank, IMP-Loss achieved 82.09% in accuracy and 79.40% in F1 score, and DIMP-Loss reached 82.67% and 79.53% respectively, while the CE-Loss reached 77.39% / 74.01%. The result showed if we use the baselines, CE-Loss, Focal-Loss, SunGen and Hu et al. (2019) to train a classifier on $D_Q$, the performance could be worse than that of using CE-Loss on much smaller $D_{P'}$ (quality checker). In contrast, our methods consistently outperformed the small quality checker, encouraging the usage of abundant LLM-generated data. Notably, even when the quality checker performs poorly, such as on the Tweet Irony dataset, where the accuracy was 62.5%, which was lower than the 76.9% achieved by directly training on generated data using CE-Loss, our methods still delivered strong performance. This suggested that a high-performance quality checker was not a prerequisite for the effectiveness of our methods. Although the performance of the meta-learning-based SunGen method was, in some cases, close to that of our methods (though still slightly below), our approaches have significant advantages in computational efficiency, making them more practical for large-scale applications. Further details on computational efficiency are shown in the Appendix E.

**IMP-Loss and DIMP-Loss Surpass the Accuracy of the Data Generator.** The GPT-3.5 few-shot predictor generated predictions using 8 examples from the small real-world dataset in the input prompt. GPT-3.5 achieved 79.46% in the Financial dataset and 68.82% in the MRPC dataset. Our approaches consistently surpassed the GPT-3.5 few-shot prediction in accuracy. The parameter size of the fine-tuned models using our methods was significantly lower than that of the GPT-3.5 data generator, yet they delivered higher performance.

**Superior and Robust Accuracy Across Epochs.** The training dynamics in Figure 1 revealed our methods outperformed CE-Loss and Focal-Loss across all benchmarks. Notably, both IMP-Loss and DIMP-Loss achieved low variation by the end of training, indicating stable performance. Moreover, DIMP-Loss showed higher variation in the initial epochs compared with IMP-Loss. This increased variability could be attributed to the order of sampled data, which caused initial fluctuations. Nevertheless, the Acc ultimately converged at a higher value than the baselines.

**Quality Checkers are Data Efficient.** Figure 2 illustrates the test accuracy on the Financial benchmark with quality checkers trained on various proportions of the original training set. As seen in this figure, even a small number of data points, e.g., 10%, was sufficient to enhance the performance of both IMP-Loss and DIMP-Loss. This suggested that a small amount of real-world data was effective for building a quality checker, making our approach efficient and practical.

---

[2]We implemented Focal-Loss, SunGen and Hu et al. (2019) by using their official code.

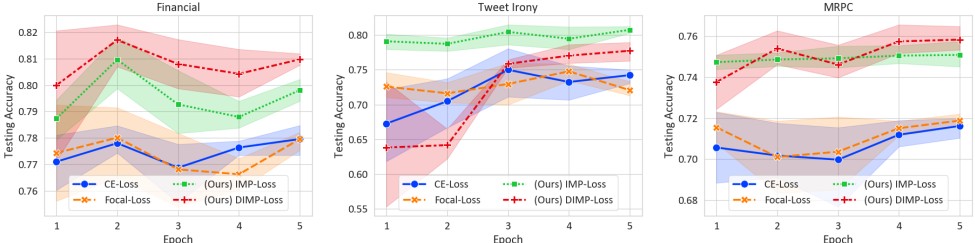

Figure 1: Training dynamics shows the testing accuracy over five epochs for benchmarks. This chart displays the minimum, maximum, and average accuracy observed across four runs with different random seeds, comparing our proposed methods with the standard CE-Loss and Focal-Loss.

| Model Size | Method | Financial | Tweet irony | MRPC |
|---|---|---|---|---|
| Base | Quality checker | 78.05 | 62.5 | 73.16 |
| Large | CE-Loss | 80.45 | 78.83 | 74.2 |
| | IMP-Loss (base DC) | 80.94 | 74.23 | 75.36 |
| | IMP-Loss (large DC) | 81.93 | 78.83 | 76.41 |
| | DIMP-Loss | **83.25** | **81.25** | **77.04** |

Table 2: Accuracy of methods on benchmarks when training a larger model with smaller Quality Checkers. "base DC" and "large DC" denote smaller and larger Diversity Checkers, respectively. Bold entries highlight the top value of metrics within each dataset.

**Diversity Checkers are Important.**    The results in Table 1 highlighted the importance of Diversity Checkers in our proposed methods. When training on GPT-3.5 generated data, the performance of the model trained with IMP-Loss without Diversity Checkers dropped compared with IMP-Loss with Diversity Checkers. For instance, in the Financial dataset, the accuracy drops from 82.09% to 81.35% and the F1 score from 79.40% to 77.94%. These results indicated that incorporating Diversity Checkers helped effectively use LLM-generated data.

**Smaller Quality Checker Still Enhances Performance by DIMP-Loss.**    The results in Table 2 illustrated the performance of each method on the benchmarks when training a larger classifier (BERT-large) with smaller Quality Checkers (BERT-base). Notably, DIMP-Loss consistently performed well even when the Quality Checker was small. This demonstrated the robustness of DIMP-Loss in adapting to different model sizes for Quality Checkers. In contrast, IMP-Loss showed inconsistent performance when using a smaller Diversity Checker compared with its training model, indicating the choice of the Diversity Checker in size significantly impacted its efficacy. In short, using a smaller Quality Checkers to guide the model was efficient in terms of both space and time.

## 5.2    Training on Real World Data

**Robust Performance of IMP-Loss and DIMP-Loss on Real-World Data.**    As shown in Table 1, IMP-Loss and DIMP-Loss outperformed other baselines even when applied directly to real-world data. Although the performance improvements are less than that of using GPT-3.5-generated data, the results indicated our methods were versatile and able to handle multiple sources of training data effectively. Specifically, in the Financial dataset, IMP-Loss achieved 85.3% Acc and 83.27% F1 score, while DIMP-Loss reached 85.4% Acc and 82.79% F1 score, surpassing CE-Loss, Focal-Loss, and (Hu et al., 2019). From our perspective, the reduced improvement in this scenario was due to the lack of a requirement to shift the training data distribution. Regarding the asymptotic viewpoint, the optimal solution of cross-entropy is already the best solution when training on real-world data. Nonetheless, our methods demonstrated robust performance across various conditions.

## 6    RELATED WORKS

We list some essential related works in this section and others in A.

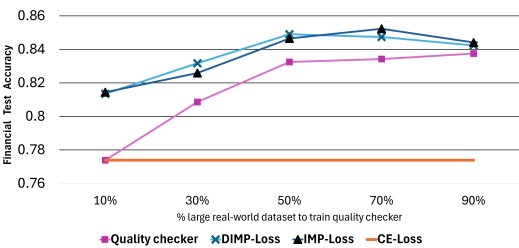

Figure 2: Test accuracy on the Financial with varying percentages of the training set for the quality checker. The graph shows the performance of each loss and the Quality Checker.

**Weighting for Misalignment Data**   Importance weighting (IW) serves as a classical strategy for addressing the issue of shifts between data distributions (Hesterberg, 1995). In traditional applications of IW, weights are derived by evaluating the degree of similarity between training and testing distributions through various statistical techniques. These techniques include maximum mean discrepancy (Schölkopf et al., 2007) and the estimation of KL divergence (Sugiyama et al., 2007). Although effective in linear model contexts, the efficacy of these methods seems to significantly diminish when applying IW to more complex deep learning frameworks (Byrd & Lipton, 2019). Besides traditional methods, recent studies have explored approaches such as Focal Loss (Lin et al., 2017) and meta-learning techniques (Hu et al., 2019; Meng et al., 2023; Gao et al., 2023), which take the weights of samples as trainable hyperparameters as discussed in Sec. 2.2.

**Synthetic Data Generation from LMs**   Recent advancements in generative AI have spurred interest in using LMs to generate synthetic data for specific tasks, particularly in low-resource settings. Studies have explored zero-shot and few-shot settings for data generation, where LMs directly generate instances with distinct labels or use little real-world data as examples to create relevant and diverse data (Li et al., 2023; West et al., 2022; Ye et al., 2022; Wang et al., 2023; Taori et al., 2023). LMs have the unique ability to generate both labels and diverse input instances, significantly enhancing the variety and quality of synthetic datasets. Approaches like ZEROGEN synthesized data by pre-trained LMs to train smaller task models, achieving competitive performance in NLP tasks, such as text classification (Ye et al., 2022).

**LM-Generated Data for Training Text Classifier**   Several studies have investigated leveraging LM-generated data for text classification. Some works maintain data quality by filtering strategy (Stylianou et al., 2023; Meng et al., 2022; 2023; Li et al., 2023; West et al., 2022; Ye et al., 2022). Another common approach is data reweighting. For example, SunGen (Gao et al., 2023) adopted a bilevel optimization approach to learn weights for synthetic data, incorporating a noise-robust loss in the outer loop to improve the reliability, and this benefited SunGen to outperform counterparts using meta-learning for data reweighting, such as Hu et al. (2019). Despite its advantages, the bilevel optimization process remains computationally expensive, making our approaches outstand by their efficiency. Moreover, there exists novel research further enhancing the use of synthetic data. For instance, UniGen (Choi et al., 2024) utilized contrastive learning to improve generalization capabilities, but required a open-source pretrained LM, while FuseGen (Zou et al., 2024) combined synthetic data from multiple LLMs to enhance performance. It is worth noting our approaches are compatible with UniGen or FuseGen, and a potential complement and enhancement of these works.

## 7   CONCLUSIONS AND DISCUSSIONS

IMP-Loss and DIMP-Loss are novel weighted-loss objectives that further enhance the performance of models trained on LLM-generated data. Our empirical results demonstrated that both methods outperformed traditional loss functions across various benchmarks. Notably, DIMP-Loss was particularly computationally efficient, requiring subtly additional resources while increasing performance. These findings emphasized the potential of IMP-Loss and DIMP-Loss in effectively leveraging synthetic data for training machine learning models. In the future, we will extend our methods on question answering, text generation, LLM pretraining, and other potential tasks, further exploring how quality and diversity matter for learning.

## 8 REPRODUCIBILITY STATEMENT

To ensure reproducibility, we provided the source code and generated dataset in supplementary materials, prompts used for generation in I.1, hyper-parameters and other training details in F, testing datasets' descriptions in H, and theoretical results in B, C, D. In addition, for the baselines, we implemented the Focal Loss as in the source code and used the publicly available code provided by Hu et al. (2019) but replaced the input data. We hope one can smoothly reproduce our results via these materials.

ACKNOWLEDGMENTS

We sincerely appreciate the insightful and valuable feedback from the anonymous reviewers. This work is primarily supported by the National Science and Technology Council, Taiwan, under grant number NSTC112-2221-E-001-025 and partially supported by E.SUN COMMERCIAL BANK. We extend our special thanks to Chien-An Chen and Yi-Ren Yeh at E.SUN for their valuable input during the preliminary discussions. Additionally, we thank the National Center for High-performance Computing (NCHC), Taiwan, for providing essential computational and storage resources.

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

## APPENDICES

## A    OTHER RELATED WORK

**Online Batch Selection**    Online batch selection (Loshchilov & Hutter, 2015; Katharopoulos & Fleuret, 2018; Mindermann et al., 2022; Deng et al., 2023) is a method to speed up training convergence by dynamically prioritizing the most informative data points, from the perspective of the minimizing loss function. This technique evaluates how informative a data point is and selects a batch $B_t$ of informative data during each training step. Unlike online batch selection methods substituting uniformly sampled batches during training, this paper focused on developing a weight function to enhance performance on downstream tasks by aligning the LLM-generated data distribution with real-world data distribution.

**Data Pruning.**    Data pruning approaches filter out noisy text data. Traditional methods took rule-based filtering for high-quality data (Bane et al., 2022; Wenzek et al., 2020), while recent approaches focused on diversification (Marion et al., 2023; Ankner et al., 2024), which used perplexity to build a diverse dataset. In advance, our methods considered both quality and diversity, and this dual focus made our weighting mechanism a possible pruning scorer. Our methods did not conflict with existing pruning methods, thus becoming a potential complement.

## B   ASYMPTOTIC CONVERGENCE OF IMP-LOSS

In this section, we provide the formal proof of the asymptotic convergence of IMP-Loss using Chebyshev's inequality. Specifically, we show that this approximately converges in probability to the expected conditional cross-entropy under $P$.

**Definition B.1 (Convergence in Probability)** *A sequence of random variables $\{X_n\}$ converges in probability to a random variable $X$, denoted as $\{X_n\} \xrightarrow{p} X$, if for any $\epsilon > 0$,*

$$\lim_{n \to \infty} P(|X_n - X| \geq \epsilon) = 0 \tag{11}$$

**Theorem B.1 (Chebyshev's Inequality)** *Let $X$ be a random variable with finite expected value $\mathbb{E}[X]$ and variance $Var(X)$. For any $\epsilon > 0$,*

$$P\left(|X - \mathbb{E}[X]| \geq \epsilon\right) \leq \frac{Var(X)}{\epsilon^2} \tag{12}$$

APPLYING CHEBYSHEV'S INEQUALITY TO IMP-LOSS

Following the definition of IMP-Loss from Eq. 6 and considering the situation without using small real-world data to approximate. Let

$$\mathcal{L}_{\text{IMP}}(\theta, D_Q) = -\frac{1}{N} \sum_{i=1}^{N} \frac{P(y_i|\mathbf{x}_i)}{Q(y_i|\mathbf{x}_i)} \log \hat{P}(y_i|\mathbf{x}_i; \theta) \tag{13}$$

Assume that all data points $(\mathbf{x}_i, y_i)$ are i.i.d. samples from the joint distribution $Q(\mathcal{X}, \mathcal{Y})$. Define

$$Z_i = -\frac{P(y_i|\mathbf{x}_i)}{Q(y_i|\mathbf{x}_i)} \log \hat{P}(y_i|\mathbf{x}_i; \theta) \tag{14}$$

The empirical mean of $Z_i$ over $N$ samples is given by:

$$\overline{Z} = \frac{1}{N} \sum_{i=1}^{N} Z_i = \mathcal{L}_{\text{IMP}}(\theta, D_Q) \tag{15}$$

The expected value of $Z_i$ under the distribution $Q$ is:

$$\mathbb{E}_Q[Z] = \mathbb{E}_Q\left[-\frac{P(y|\mathbf{x})}{Q(y|\mathbf{x})} \log \hat{P}(y|\mathbf{x}; \theta)\right] \tag{16}$$

Applying Chebyshev's inequality to the sequence $\overline{Z}$:

$$P\left(|\overline{Z} - \mathbb{E}_Q[Z]| \geq \epsilon\right) \leq \frac{\text{Var}_Q(\overline{Z})}{\epsilon^2} = \frac{\text{Var}_Q(Z_1)}{N\epsilon^2} \tag{17}$$

As $N$ grows large, the right-hand side converges to zero, implying that $\overline{Z}$ converges in probability to $\mathbb{E}_Q[Z]$.

Therefore,

$$\mathcal{L}_{\text{IMP}}(\theta, D_Q) \xrightarrow{p} \mathbb{E}_Q\left[-\frac{P(y|\mathbf{x})}{Q(y|\mathbf{x})} \log \hat{P}(y|\mathbf{x}; \theta)\right] \tag{18}$$

TRANSFORMING FROM $Q$ TO $P$

Next, we show that:

$$
\begin{aligned}
& \mathbb{E}_Q\left[-\frac{P(y|\mathbf{x})}{Q(y|\mathbf{x})}\log \hat{P}(y|\mathbf{x};\theta)\right] \\
& = -\sum_{\mathbf{x}\in\mathcal{X}}\sum_{y\in\mathcal{Y}}Q(\mathbf{x},y)\frac{P(y|\mathbf{x})}{Q(y|\mathbf{x})}\log \hat{P}(y|\mathbf{x};\theta) \\
& = -\sum_{\mathbf{x}\in\mathcal{X}}Q(\mathbf{x})\sum_{y\in\mathcal{Y}}P(y|\mathbf{x})\log \hat{P}(y|\mathbf{x};\theta) \\
& \approx -\sum_{\mathbf{x}\in\mathcal{X}}P(\mathbf{x})\sum_{y\in\mathcal{Y}}P(y|\mathbf{x})\log \hat{P}(y|\mathbf{x};\theta) \\
& = \mathbb{E}_P\left[-\log \hat{P}(y|\mathbf{x};\theta)\right]
\end{aligned}
\tag{19}
$$

Given that $Q(\mathbf{x}) \approx P(\mathbf{x})$ by the assumption that the LLM is capable of simulating the real-world input distribution through careful and appropriate prompting, we have:

$$
\begin{aligned}
& \mathcal{L}_{\text{IMP}}(\theta, D_Q) \\
& \xrightarrow{p} \mathbb{E}_Q\left[-\frac{P(y|\mathbf{x})}{Q(y|\mathbf{x})}\log \hat{P}(y|\mathbf{x};\theta)\right] \\
& \approx \mathbb{E}_P\left[-\log \hat{P}(y|\mathbf{x};\theta)\right]
\end{aligned}
\tag{20}
$$

Thus, the asymptotic convergence of IMP-Loss ensures that the weighted loss function effectively aligns the LLM-generated data distribution $Q$ with the real-world data distribution $P$.

## C    DERIVATION OF DIMP-LOSS

In this section, we provide the formal derivation to address the question: *Which data point in $D_Q$, when used for training, will most effectively bring the model distribution closest to $P$?* Following the optimization formulation in Eq. 7, we can empirically apply Monte Carlo estimation using a small real-world dataset $D_{P'}$, denoted as $(\mathbf{y}_{P'}, \mathbf{X}_{P'})$. This allows us to reformulate the problem by maximizing the joint probability of the data points in $D_{P'}$, which leads to the following optimization problem. This derivation is similar to the online batch selection techniques discussed in previous research (Deng et al., 2023).

$$
\underset{(\mathbf{x},y)\in D_Q}{\arg\max} \hat{P}(\mathbf{y}_{P'}|\mathbf{X}_{P'};\theta_t,\{(\mathbf{x},y)\}) = \underset{(\mathbf{x},y)\in D_Q}{\arg\max} \prod_{(\mathbf{x}',y')\in D_{P'}} \hat{P}(y'|\mathbf{x}';\theta_t,\{(\mathbf{x},y)\})
\tag{21}
$$

This formulation leverages the joint probability of the dataset $D_{P'}$, ensuring that the selected data points in $D_Q$ are those that, when used for training, most effectively align the model's distribution with the small real-world distribution $P'$. This also implies that the chosen data point leads the model to perform well on $D_{P'}$, enhancing the likelihood of better generalization to real-world data.

APPLYING BAYES RULE

By applying Bayes' rule to the joint probability of the dataset $D_{P'}$, we obtain:

$$\hat{P}(\mathbf{y}_{P'}|\mathbf{X}_{P'}; \theta_t, \{(\mathbf{x}, y)\})$$

$$= \frac{\hat{P}(D_{P'}, \mathbf{x}, y, \theta_t)}{\hat{P}(\mathbf{X}_{P'}, \mathbf{x}, y, \theta_t)}$$

$$= \frac{\hat{P}(\mathbf{y}_{P'}|\mathbf{X}_{P'}, \mathbf{x}, \theta_t)\hat{P}(y|\mathbf{x}, D_{P'}, \theta_t)}{\hat{P}(y|\mathbf{x}, \mathbf{X}_{P'}}, \theta_t) \tag{22}$$

$$= \frac{\hat{P}(\mathbf{y}_{P'}|\mathbf{X}_{P'}, \theta_t)\hat{P}(y|\mathbf{x}; D_{P'}, \theta_t)}{\hat{P}(y|\mathbf{x}; \theta_t)}$$

The final equality holds because $\mathbf{x}$ alone cannot perform a model update, leading to the conditional independence assumption. Since $\hat{P}(\mathbf{y}_{P'}|\mathbf{x}_{P'}, \mathbf{x}, \theta_t)$ is a constant for this optimization problem and does not influence the result, we can further simplify the optimization as follows:

$$\underset{(\mathbf{x},y)\in D_Q}{\arg\max} \hat{P}(\mathbf{y}_{P'}|\mathbf{X}_{P'}; \theta_t, \{(\mathbf{x}, y)\}) =$$

$$\underset{(\mathbf{x},y)\in D_Q}{\arg\max} \frac{\hat{P}(y|\mathbf{x}; \theta_t, D_{P'})}{\hat{P}(y|\mathbf{x}; \theta_t)} \tag{23}$$

Similar to the online batch selection work, we use $P'(y|\mathbf{x})$ to approximate $\hat{P}(y|\mathbf{x}; \theta_t, D_{P'})$. This approximation is then utilized as the weight in our loss function. Consequently, if a data point brings the model closer to the real-world distribution $P$, its corresponding weight will be higher, thus impacting the model's training.

## D    LOWER BOUND OF DIMP-LOSS

**The Lower Bound**

$$N\mathcal{L}_{\text{DIMP}}(\theta_t, D_Q) = -\sum_{i=1}^{N} \frac{\hat{P}'(y_i|\mathbf{x}_i)}{\hat{P}(y_i|\mathbf{x}_i; \theta_t)} \log \hat{P}(y_i|\mathbf{x}_i; \theta_t)$$

$$\text{(By *AM-GM inequality: } \frac{1}{a} \geq 2 - a \text{ for } a = \hat{P}(y_i|\mathbf{x}_i; \theta_t))$$

$$\geq -\sum_{i=1}^{N} \hat{P}'(y_i|\mathbf{x}_i) \left(2 - \hat{P}(y_i|\mathbf{x}_i; \theta_t)\right) \log \hat{P}(y_i|\mathbf{x}_i; \theta_t)$$

$$= -2\sum_{i=1}^{N} \hat{P}'(y_i|\mathbf{x}_i) \log \hat{P}(y_i|\mathbf{x}_i; \theta_t) - \left|\sum_{i=1}^{N} \hat{P}'(y_i|\mathbf{x}_i)\hat{P}(y_i|\mathbf{x}_i; \theta_t) \log \hat{P}(y_i|\mathbf{x}_i; \theta_t)\right|$$

$$\text{(By Hölder's Inequality } \|fg\|_1 \leq \|f\|_\infty \|g\|_1)$$

$$\geq -2\sum_{i=1}^{N} \hat{P}'(y_i|\mathbf{x}_i) \log \hat{P}(y_i|\mathbf{x}_i; \theta_t) - \max_i \hat{P}'(y_i|\mathbf{x}_i) \sum_{i=1}^{N} \left|\hat{P}(y_i|\mathbf{x}_i; \theta_t) \log \hat{P}(y_i|\mathbf{x}_i; \theta_t)\right|$$

$$\geq \underbrace{-\sum_{i=1}^{N} \hat{P}'(y_i|\mathbf{x}_i) \log \hat{P}(y_i|\mathbf{x}_i; \theta_t)}_{\text{Empirical distilled cross-entropy loss}} + \underbrace{\sum_{i=1}^{N} \hat{P}(y_i|\mathbf{x}_i; \theta_t) \log \hat{P}(y_i|\mathbf{x}_i; \theta_t)}_{\text{Maximum entropy regularizer}}$$

$$\tag{24}$$

**\*AM-GM Inequality**    The derivation shown illustrates the application of the **Arithmetic Mean - Geometric Mean (AM-GM) inequality**, which states that for any two positive numbers $x$ and $y$, the arithmetic mean is greater than or equal to the geometric mean, i.e., $\frac{a+b}{2} \geq \sqrt{ab}, \quad \forall a, b > 0$. In this specific case, $b$ is set to $\frac{1}{a}$, simplifying the inequality to:

$$\frac{a + \frac{1}{a}}{2} \geq \sqrt{1} = 1.$$

Multiplying both sides by 2 yields:

$$a + \frac{1}{a} \geq 2,$$

and rearranging the inequality gives:

$$\frac{1}{a} \geq 2 - a.$$

This result is a classic application of the AM-GM inequality, demonstrating that the sum of a number and its reciprocal is always greater than or equal to 2 for any positive $x$.

# E COMPUTATIONAL TIME COMPARISON

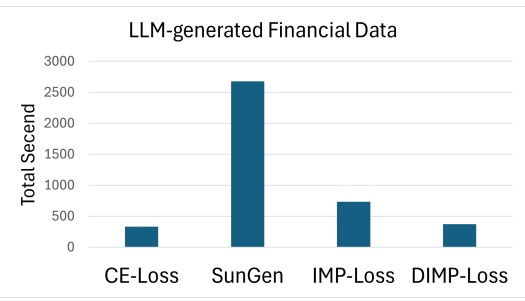

Figure 3: Total running time (in seconds) for CE-Loss, IMP-Loss, and DIMP-Loss on the LLM-generated Financial benchmark.

| Method | Build QC | Build DC | Precalculate weights | Training | Total |
|--------|----------|----------|----------------------|----------|-------|
| CE-Loss | - | - | - | 333.242s | 333.242s |
| SunGen | - | - | - | 2680s | 2680s |
| IMP-Loss | 8.824s | 333.516s | 57.695s | 333.328s | 733.363s |
| DIMP-Loss | 8.824s | - | 29.274s | 333.426s | 371.524s |

Table 3: Total running time of each component (in seconds) for CE-Loss, SunGen, IMP-Loss, and DIMP-Loss on the LLM-generated financial benchmark. The table breaks down the time spent building the Quality Checker (QC), building the Diversity Checker (DC), precalculating weights, and training. The total time combines all these components.

In this computational time experiment, we evaluated the running times on the LLM-generated Financial benchmark dataset, which includes 10,012 training samples ($D_Q$) and 242 development samples (small real-world data, $D_{P'}$). Our comparison focused on four methods: CE-Loss, SunGen, IMP-Loss, and DIMP-Loss. We have broken down the total process into four components: building the Quality Checker (QC), building the Diversity Checker (DC), precalculating constant weights, and the actual training time for each respective loss function. For all experiments, the downstream models and checkers were trained for 5 epochs with a batch size of 32. The batch size was set to 64 during the precalculating constant weights phase. The inner loop epochs were set to 1 for SunGen. The results of this breakdown are presented in Table 3, and the total time is in Figure 3. The results indicate that IMP-Loss requires approximately 2.2 times the running time of CE-Loss. In contrast, This demonstrates that DIMP-Loss is highly efficient, requiring only a slight overhead compared to CE-Loss, while SunGen's computational time is approximately 7 times higher, further underscoring the efficiency of our methods for large-scale applications.

# F TRAINING DETAILS AND HYPERPARAMETERS

For our experiments, we used a pre-trained BERT-base model (Devlin et al., 2019) from Hugging-face's transformers library (Wolf et al., 2020) as the encoder, utilizing the representation embedding from the last layer as input to our classification models. We fine-tuned the model with hyperparameters selected from the following ranges: learning rate {6e-6, 6e-5}, epochs {5, 7}, and batch size

{32, 64}. Other hyperparameters were set to the default values provided by Huggingface's trainer for text classification. The best checkpoint was selected based on the accuracy of the development set. We repeated each experiment with four random seeds. We reported the best accuracy run on tables (Table 1), while also providing the minimum, maximum, and average in the training dynamics section (Sec. 5.1). To train the quality checker, we used the small real-world dataset (development split) not included in the training data and trained the quality checker for five epochs. Similarly, the diversity checker of IMP-Loss was also trained for five epochs. This approach aligns with our setup, where access to real-world data is limited, and thus, we reuse the development set to build the quality checker and perform model selection. All experiments were conducted using PyTorch (Paszke et al., 2019) and Huggingface (for models and datasets) on V100 GPUs with 32GB memory.

## G    TRAINING ON NOISY DATA

| Dataset | Method | Financial | | Tweet Irony | | MRPC | |
|---|---|---|---|---|---|---|---|
| | | Acc | F1 | Acc | F1 | Acc | F1 |
| Small real world | GPT-3.5 few-shot | 79.46 | 81.6 | 63.39 | 69.39 | 69.28 | 71.75 |
| | CE-Loss (quality checker) | 78.05 | 75.26 | 62.5 | 62.38 | 73.16 | 68.69 |
| Noisy Data | CE-Loss | 78.38 | 73.44 | 60.46 | 60.14 | 74.03 | 67.5 |
| | Focal-Loss | 78.55 | 74.97 | 62.11 | 61.12 | 74.72 | 69.59 |
| | IMP-Loss (Ours) | 81.6 | 78.24 | **64.8** | **64.51** | 76 | 70.46 |
| | DIMP-Loss (Ours) | **82.59** | **80.28** | 64.16 | **64.09** | **76.58** | **71.32** |

Table 4: Performance metrics on the noisy data. The table showcases the accuracy (Acc) and macro F1 score (F1) for each method applied on three distinct datasets: Financial, Tweet Irony, and MRPC. The methods include CE-Loss, GPT-3.5 few-shot, Focal-Loss, IMP-Loss, and DIMP-Loss. Notably, bold entries indicate the best-performing metrics within each training dataset category.

In this section, we evaluate the robustness of our proposed methods, IMP-Loss and DIMP-Loss, by training on noisy datasets. We aim to simulate real-world scenarios where LLM-generated data may be imperfect due to labeling errors (low quality), duplicate entries (low diversity), and unrelated inputs (low quality). This allows us to analyze the effects of the Quality Checker and Diversity Checker in IMP-Loss and DIMP-Loss.

### G.1    EXPERIMENTAL SETUP

To create noisy datasets, we start with the original training set from each benchmark (Financial, Tweet Irony, and MRPC) and split it into three parts:

1. **Original Data:** This part remains unchanged and serves as the control set.

2. **Random Swapped Label Noise:** In this part, the labels are randomly altered, introducing label noise and reducing data quality.

3. **Duplicated Data:** In this part, each data point is duplicated once, introducing redundancy and reducing data diversity.

4. **Unrelated Input Data (Only for Financial):** For the financial benchmark, we introduce out-of-domain input noise by randomly selecting 452 data points from the Tweet Sentiment Extraction benchmark (Maggie et al., 2020). While this dataset is also a sentiment classification task, it is unrelated to the financial domain.

### G.2    PERFORMANCE RESULTS

The results in Tabel 4 indicated that our proposed methods, IMP-Loss and DIMP-Loss, consistently outperform the baselines across all benchmarks, even when the training data is noisy. Specifically, in the Financial dataset, IMP-Loss achieves 81.6% Acc and 78.24% F1 score, while DIMP-Loss reaches 82.59% Acc and 80.28% F1 score, surpassing the CE-Loss and Focal-Loss baselines. In the Tweet Irony dataset, the performance improvement is more pronounced, with IMP-Loss achieving 64.8% Acc and 64.51% F1 score, and DIMP-Loss achieving 64.16% Acc and 64.09% F1 score,

significantly higher than CE-Loss and Focal-Loss. For the MRPC dataset, IMP-Loss and DIMP-Loss show robust performance with 76% Acc and 70.46% F1 score and 76.58% Acc and 71.32% F1 score, respectively, outperforming the GPT-3.5 few-shot approach, which achieves 69.28% Acc and 71.75% F1 score.

### G.3 ANALYSIS OF CHECKER SCORES AND WEIGHTS

**IMP-Loss**  Figure 4 and Figure 5 illustrate the Quality Checker Score $P'(y|\mathbf{x})$, Diversity Checker Score $Q(y|\mathbf{x})$, and the corresponding weights of the IMP-Loss for the Financial and the Tweet Irony dataset, across three different data conditions: original, swapped labels, and duplicated entries.

The Quality Checker Score is highest for the original data and significantly lower for the swapped label data, indicating that the model correctly identifies the labels as lower quality. The Diversity Checker Score (where lower values are better) is lower for the original data than the duplicated data, indicating the impact of duplication on diversity. Additionally, the swapped label data achieves the highest diversity because the altered labels create data points that are substantially distinct from the rest of the dataset. Similarly, the unrelated input data exhibits relatively high diversity due to its out-of-domain nature. However, data points from both the swapped label and unrelated input categories have low Quality Checker Scores, resulting in their lower assigned weights. Consequently, the weights assigned to the original data are higher compared to the swapped label data and the duplicated data, demonstrating the effectiveness of IMP-Loss in recognizing and appropriately weighting high-quality, diverse data.

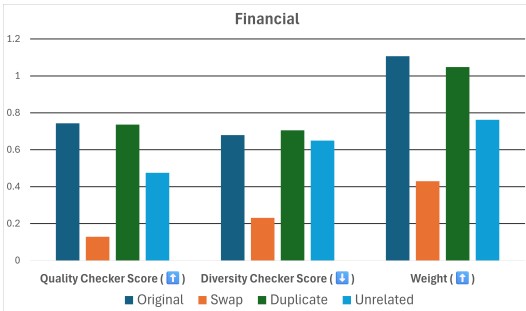

Figure 4: Average Quality Checker Score, Diversity Checker Score, and Weights of IMP-Loss for Financial Dataset: Comparison between Original, Swapped Label, Duplicated Data and Unrelated Input Data.

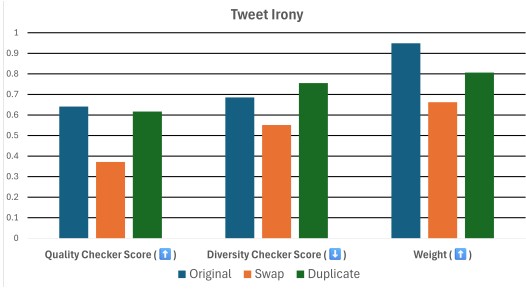

Figure 5: Average Quality Checker Score, Diversity Checker Score, and Weights of IMP-Loss for Tweet Irony Dataset: Comparison between Original, Swapped Label, and Duplicated Data

**DIMP-Loss**  In contrast, Figure 6 shows the diversity scores $\hat{P}(y|\mathbf{x};\theta_t)$ and the weights for the DIMP-Loss method on Financial benchmark across epoch, where the diversity checker is the training model itself. The Diversity Checker Score (lower is better) is also lower for the original data than the duplicated data and Unrelated input data. In the end, The weights (Figure 7) assigned by the DIMP-Loss method are consistently higher for the original data than the swapped label, Unrelated

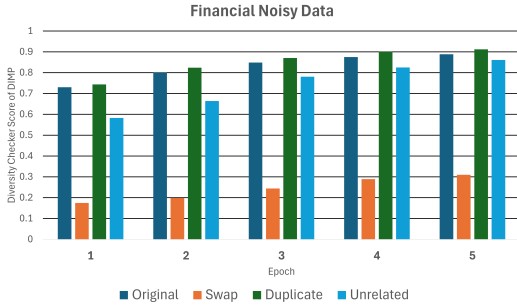

Figure 6: Average Diversity Checker Score of DIMP-Loss for Original, Swapped label, Unrelated input data, and Duplicated data on the Financial Dataset across Epoch.

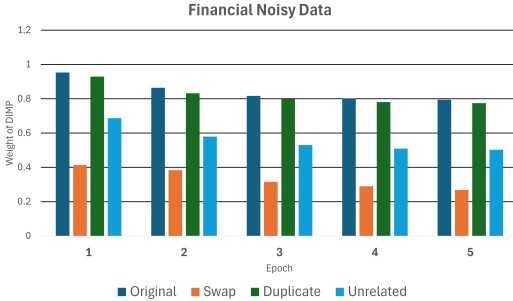

Figure 7: DIMP-Loss Weight for Original, Swapped label, Unrelated input data, and Duplicated Data on the Financial Dataset across Epoch.

input data, and duplicated data across epochs. This pattern aligns with the results observed for the IMP-Loss method.

## H  DATASET DESCRIPTIONS

The size of each split and the generated data in Table 5.

|           | Financial | Tweet Irony | MRPC  |
|-----------|-----------|-------------|-------|
| Train     | 3392      | 2862        | 3668  |
| Dev       | 242       | 200         | 408   |
| Test      | 1212      | 784         | 1,725 |
| Generated | 10012     | 3000        | 3005  |

Table 5: Data size of each split

The description of each dataset is following:

**Financial Phrasebank:**    This benchmark involves categorizing finance-related sentences into positive, negative, or neutral sentiment categories. These sentences, numbering 4,840, are extracted from financial news articles. Since the dataset does not come with predefined training, validation, and testing splits, we randomly divided it into training (70%), validation (5%), and testing (25%) sets like the previous work (Li et al., 2023).

**Tweet Irony:**    This task requires sorting tweets into two groups: ironic and non-ironic. The dataset containing tweets in English has been explicitly annotated for these categories. It comprises 2,862 instances for training and 784 instances for testing. Initially, there were 955 instances in the validation set, but due to limited access to real-world data in our scenario, we have randomly selected 200 instances for our validation sets.

| Task | Prompt |
|------|--------|
| Tweet irony | **System Prompt:** Now you are a person using Twitter. You are asked to write an irony or non-irony tweet to express your feelings. Your writing style must be consistent with the texts in the tweet. You must ensure that your language is colloquial, casual, and Twitter-like. You are given a length requirement. You must ensure your tweet meets the length requirement. |
| | **Data Generation Prompt:** Write a tweet expressing {label} feeling and ensure that the length of the tweet is about {num_of_words} words. Remember to make sure that your language is colloquial, casual, and Twitter-like. Be creative and write unique tweets.
For example:
{Examples of the label from small-real world dataset}...
Can you provide something more diverse than the previously generated data? |
| Financial | **Context Prompt:** You are now a journalist writing financial news. You need to write some financial news that expresses polar sentiments. The financial news you generate needs to be considered from an investor's viewpoint only, i.e., whether the news may have a positive, negative, or neutral influence on the stock price. As a result, sentences with a sentiment irrelevant from an economic or financial perspective are considered neutral. You are given one of the polar sentiments and a length requirement. You must write financial news that expresses the corresponding sentiment and meets the length requirement. |
| | **Data Generation Prompt:** Write financial news with {label} sentiment and ensure that the length of the financial news is about {num_of_words} words. Be creative and write unique financial news.
For example:
{Examples of the label from small-real world dataset}...
Can you provide something more diverse than the previously generated data? |
| MRPC | **Context Prompt:** Generate {num_of_examples} data points like the following examples. A label of 1 means they are semantically similar, and a label of 0 means they are not. Try to balance the number of each category (Please just output the format like what I provide, and the output MUST be different from input): |
| | **Data Generation Prompt:**
For example:
sentence1: Amrozi accused his brother, wh—om he called " the witness ", of deliberately distorting his evidence .—— sentence2: Referring to him as only " the witness ", Amrozi accused his brother of deliberately distorting his evidence .—— label: 1
sentence1: They had published an advertisement on the Internet on June 10, offering the cargo for sale, he added .—— sentence2: On June 10, the ship's owners had published an advertisement on the Internet, offering the explosives for sale. —— label: 1
{Other examples from small-real world dataset}...
Can you provide something more diverse than the previously generated data? |

Table 6: Detailed prompts for each task for data generation.

**MRPC:** The Microsoft Research Paraphrase Corpus (MRPC) consists of 5,801 sentence pairs sourced from news articles. Human annotators manually labeled each pair to determine whether the sentences were paraphrased from each other. We employ the official MRPC dataset available through Huggingface's datasets library, segmented into training, validation, and testing sets containing 3,668, 408, and 1,725 instances, respectively.

# I  DATA GENERATION

## I.1  PROMPT

The prompts used for data generation across different benchmarks are provided in Table 6. The prompts for Tweet Irony and Financial datasets are based on those used in previous work (Li et al., 2023).

## I.2  DATA GENERATION BUDGET

We used OpenAI GPT-3.5-turbo-1106 (OpenAI, 2022) to generate a dataset for the three benchmarks, adhering to OpenAI's terms of service and usage policies. The total cost is $38.74.

