# OpenReview forum: "Not All LLM-Generated Data Are Equal: Rethinking Data Weighting in Text Classification"
_ICLR.cc/2025/Conference — ICLR 2025 Spotlight_

### Official Review · Reviewer_e1Ke · 2024-10-22

**Soundness:** 3
**Presentation:** 4
**Contribution:** 3
**Rating:** 8
**Confidence:** 5

**Summary:**

This paper proposes a modification to the loss function to improve the performance of models trained on synthetic data, making it comparable to models trained on real-world data. This approach addresses the challenge of limited availability of real-world data, where models trained solely on synthetic data often underperform. The authors introduce two new loss functions—IMP-Loss and DIMP-Loss—which incorporate mechanisms for quality-checking and diversity-checking the synthetic data. IMP-Loss prioritizes synthetic samples that closely resemble real-world distributions, while DIMP-Loss emphasizes samples that guide the model toward the performance of a small base model trained on limited real-world data. Experimental results demonstrate that these loss functions enhance the performance of models trained on synthetic data, outperforming traditional baseline methods.

**Strengths:**

Overall, this paper is well-written, addressing a compelling and practical problem for researchers working with synthetic data. The issue of model performance degradation when using synthetic data is aligned with recent research findings. The proposed loss functions IMP-Loss and DIMP-Loss are creative and effective solutions.

The clarity of the paper is a strong point, with a well-defined problem formulation and a meticulous derivation of the proposed loss functions. Each step in the reasoning process is clearly outlined, making the theoretical contributions easy to follow.

The paper includes extensive experimentation, which demonstrates the efficacy of the proposed methods. The analysis of the runtime complexity further adds the practical applicability of the algorithm in real-world scenarios.

**Weaknesses:**

The core idea of weighting synthetic data samples is not entirely novel. A similar concept was explored in the paper "Self-Guided Noise-Free Data Generation for Efficient Zero-Shot Learning" published last year. Although the formulation and implementation in this paper differ, the underlying principle of prioritizing certain synthetic data points is conceptually similar, which somewhat limits the novelty of the contribution.

Another concern lies in the strong reliance on the assumption that a model trained on a small real-world dataset P can approximate the performance of a model trained on a larger, complete dataset. The paper would benefit from a more in-depth discussion or experimentation addressing this assumption. Specifically, it would be helpful to explore the minimum number of real-world examples required for this assumption to hold and to provide guidelines on the amount of data practitioners should aim to collect.

**Questions:**

In some formulas, it's unclear whether the pairs (xi, yi) are drawn from the real-world distribution P or the synthetic data distribution Q. For instance, in Appendix C, it is clearly noted that (yp', xp') is from distribution P. Could you use similar notation consistently throughout the main body to help readers differentiate between real and synthetic data sources?

To improve clarity, consider starting a new paragraph with the sentence: "Thus, inspired by the objective of Hu et al. (2019), we introduce two novel, efficient, automatic weighted-loss approaches: Importance Loss (IMP-Loss) and Dynamic Importance Loss (DIMP-Loss)..." This will make the contributions of this paper stand out more clearly, preventing them from being mixed with discussions of prior work.

---

> ### Author Response · Authors · 2024-11-20
> **Reply to Reviewer e1Ke**
>
> We sincerely thank Reviewer e1Ke for reviewing our submission and for recognizing the clarity, creativity, and practical value of our work. Below, we address your comments in detail.
>
> ## Reply to Q1
> > In some formulas, it's unclear whether the pairs (xi, yi) are drawn from the real-world distribution P or the synthetic data distribution Q. For instance, in Appendix C, it is clearly noted that (yp', xp') is from distribution P. Could you use similar notation consistently throughout the main body to help readers differentiate between real and synthetic data sources?
>
> Thank you for pointing out this potential issue regarding the notation. In Appx C of this version, $(\mathbf{y}_p', \mathbf{x}_p')$ represents a dataset, where $\mathbf{y}_p'$ is a set of ground truth labels and $\mathbf{x}_p'$ is a set of input data. We found this notation may have caused some confusion, since $(\mathbf{y}_p', \mathbf{x}_p')$ looks like a data point. Therefore, to improve clarity, Eq(21) should be interpreted as the prodcut of each point's conditional probability (due to i.i.d):
>
> $$
> \arg\max_{(\mathbf{x}, y) \in D_Q} \prod_{(\mathbf{x}', y') \in D_{P'}} \hat{P}(y'|\mathbf{x}'; \theta_{t}, \{(\mathbf{x}, y)\})
> $$
> We are revising the terms throughout the manuscript. For instance, the confusing $(\mathbf{y}_p', \mathbf{x}_p')$ could be updated to $(\mathbf{y}_p', \mathbf{X}_p')$ to clearly indicate this refers to a dataset, while variables such as $(\mathbf{x}_i, y_i)$ from $D_Q$ could be $(\mathbf{x}_i^{Q}, y_i^{Q})$, and similarly for other distributions. We think this update would help differentiate between real and synthetic data sources more consistently and intuitively.
>
> We appreciate your suggestion, and we will refine our notations in the next revision as soon as possilbe to ensure the notation is  unambiguous.
> ## Reply to Q2
> > To improve clarity, consider starting a new paragraph with the sentence: "Thus, inspired by the objective of Hu et al. (2019), we introduce two novel, efficient, automatic weighted-loss approaches: Importance Loss (IMP-Loss) and Dynamic Importance Loss (DIMP-Loss)..." This will make the contributions of this paper stand out more clearly, preventing them from being mixed with discussions of prior work.
>
> Thank you for carefully reviewing our work and providing this valuable suggestion. We have incorporated your feedback and made the necessary edits in the revised version of the manuscript.
> ## Reply to Weaknesse1
> > The core idea of weighting synthetic data samples is not entirely novel. A similar concept was explored in the paper "Self-Guided Noise-Free Data Generation for Efficient Zero-Shot Learning" published last year. Although the formulation and implementation in this paper differ, the underlying principle of prioritizing certain synthetic data points is conceptually similar, which somewhat limits the novelty of the contribution.
>
> Thank you for highlighting this impressive work. In the revised manuscript, we have extended our experiments to include a comparison with SunGen [1], which is closely related to the idea of weighting synthetic data samples. The results demonstrate in Table 1 and Table 3, shows that while SunGen employs a meta-learning-based approach that requires approximately seven times the computational cost of our DIMP-Loss method, our approach still consistently outperforms SunGen in overall performance. These findings suggest that our method offers a promising and more efficient alternative for advancing the field of synthetic data weighting. We believe this highlights the potential of our approach as a practical and impactful contribution to this area of research.
>
> ## Reply to Weaknesse2
> > Another concern lies in the strong reliance on the assumption that a model trained on a small real-world dataset P can approximate the performance of a model trained on a larger, complete dataset. The paper would benefit from a more in-depth discussion or experimentation addressing this assumption. Specifically, it would be helpful to explore the minimum number of real-world examples required for this assumption to hold and to provide guidelines on the amount of data practitioners should aim to collect.
>
> We appreciate your concern regarding the reliance on a small real-world dataset to approximate the larger distribution. In response to your feedback, we have included CE-Loss in Figure 2 in the revised manuscript to facilitate easier comparisons. Empirically, we provided a comparison of different amounts of training data used for the quality checker, as illustrated in Figure 2. The results indicate that increasing the size of the real-world training dataset improves performance. However, even a small amount of real-world data proved sufficient to significantly enhance the performance of both IMP-Loss and DIMP-Loss compared to directly using CE-Loss and the Quality Checker alone.
>
> ## Reference
> [1] Gao et al., Self-Guided Noise-Free Data Generation for Efficient Zero-Shot Learning, ICLR 2023.

---

> > ### Comment · Reviewer_e1Ke · 2024-11-26
> > **To Authors**
> >
> > Thank you for your rebuttal. I was satisfied with the revisions made by the authors and I don't have further questions.

---

### Official Review · Reviewer_eiLk · 2024-10-28

**Soundness:** 4
**Presentation:** 3
**Contribution:** 3
**Rating:** 8
**Confidence:** 3

**Summary:**

This paper proposes a reweighting scheme for synthetic data generated by LLMs. The focus is on the distinction between the distributions of real and synthetic data. To address this, the authors suggest IMP-Loss and DIMP-Loss to align the distribution of synthetic data with that of real data. These two loss functions assign weights to the cross-entropy loss for each data point. The empirical results support the effectiveness of this approach.

(Note: my review has been revised by LLM for improving the grammar.)

**Strengths:**

- The use of synthetic data has been of significant interest to researchers for years. It is also important to note that noise in synthetic data has been a long-standing concern. While several studies have proposed methods to address this issue, they come with their own limitations, as the current manuscript notes. This work is timely in addressing those previous limitations.
  - While the proposed method requires some additional computation for assigning weights, I agree that it is more efficient compared to methods based on meta-learning, at least.
  - As a result, this method seems more practical than previous approaches for mitigating the noise in synthetic data.

**Weaknesses:**

- **Most of my concerns are resolved during the rebuttal process.**

- ~~I see several issues with the introduction and related work section.~~
  - The authors primarily build their argument by citing studies focused on generating synthetic data for *instruction tuning of LLMs*, such as Alpaca.
  - However, the method proposed in this paper is aimed at mitigating noise in synthetic data for *text classification*, which might create some confusion.
  - The confusion arises because there are many studies focused on generating synthetic data specifically for text classification, which are more relevant to this work, but the authors do not cite them.
    - Meng et al., Generating Training Data with Language Models: Towards Zero-Shot Language Understanding, NeurIPS 2022.
    - Meng et al., Tuning Language Models as Training Data Generators for Augmentation-Enhanced Few-Shot Learning, ICML 2023.
    - Gao et al., Self-Guided Noise-Free Data Generation for Efficient Zero-Shot Learning, ICLR 2023.
    - Choi et al., UniGen: Universal Domain Generalization for Sentiment Classification via Zero-Shot Dataset Generation, arXiv Preprint 2024.
    - Zou et al., FuseGen: PLM Fusion for Data-Generation-Based Zero-Shot Learning, arXiv Preprint 2024.
  - I believe discussing these studies would be more relevant compared to the current organization of the manuscript.
- ~~Accordingly, the authors should have discussed these works and used them as baselines.~~
  - For instance, SunGen (Gao et al., 2023), UniGen, and FuseGen propose methods for mitigating the noise in synthetic data.
  - SunGen, in particular, is very similar to this work, as it suggests reweighting synthetic data based on quality. While SunGen is based on meta-learning and may require more computation than the proposed method, the current organization of the manuscript does not adequately address this overlap.
  - Furthermore, FuseGen suggests a reweighting scheme that does not require additional computation, unlike the proposed method.

~~Given these points, I do not believe this paper is ready for publication at ICLR in its current form.~~

**Questions:**

- I am still uncertain whether the proposed diversity checker functions as intended. What happens if unrelated data is fed into the diversity checker? SunGen has already demonstrated that noise in synthetic data is not limited to data with noisy labels but also includes data unrelated to the desired domain.
- Some comments regarding the paper's formatting:
  - Line 030: Use a capital letter for the abbreviation of LLM. Currently, it reads 'Large language models,' which is inconsistent with other terms like 'Natural Language Processing.'
  - Line 063: Ensure proper spacing between ‘ers’ and ‘highly.’
  - Footnote 1: Start the sentence with a capital letter, and ensure there is a period at the end.
  - Please ensure consistent use of \citet and \citep, as this inconsistency currently reduces the overall quality of the paper. For example:
    - Line 033: Use \citet for Taori et al.
    - Line 050: Use \citet for Hu et al.

---

> ### Author Response · Authors · 2024-11-20
> **Reply to Reviewer eiLk (Part 1/2)**
>
> We sincerely thank Reviewer eiLk for taking the time to review our submission and providing thoughtful feedback. We are grateful for your recognition of our work's potential and the strong positive results it demonstrates. Below, we provide detailed responses to each of your comments.
> # Reply to Q1
> > I am still uncertain whether the proposed diversity checker functions as intended. What happens if unrelated data is fed into the diversity checker?
>
> Thank you for your insightful suggestion. To address this concern, we have extended our noise data experiments to this revision in Appendix G by incorporating unrelated input data from a financial benchmark. The average weights derived from the Quality Checker, Diversity Checker, and overall weight adjustments (Figures 4, 6, and 7) provide a comprehensive analysis. Our findings indicate that while the Diversity Checker assigns slightly lower scores to unrelated data (lower is better), the Quality Checker scores drop significantly (higher is better). This discrepancy results in a considerable reduction in the overall weight of unrelated data compared to both the original data and duplicate data. These findings align with our derivations, emphasizing that the model should be trained on data points that exhibit both quality and diversity properties.
>
> # Reply to Q2
> > Some comments regarding the paper's formatting
>
> Thank you for the suggestions. We have double checked the formatting of capitalization, spacing and citations.
>
> # Reply to Weakness 1
> > The authors primarily build their argument by citing studies focused on generating synthetic data for instruction tuning of LLMs, such as Alpaca. However, the method proposed in this paper is aimed at mitigating noise in synthetic data for text classification, which might create some confusion.
>
> Thank you for pointing out these potential issues in our paper. We appreciate your suggestion regarding the relevance of related studies. In response, we have revised the Introduction and Related Work sections to highlight the studies you mentioned, which are indeed more pertinent to our proposed methods. These updates are clearly marked in red in the latest version of the manuscript to ensure they are easily identifiable. Your feedback has helped us improve the clarity and relevance of our target task.
>
> # Reply to Weakness 2
> > I believe discussing these studies would be more relevant compared to the current organization of the manuscript. Accordingly, the authors should have discussed these works and used them as baselines.
>
> Thank you for your valuable feedback and suggestions. To address your concerns and improve the manuscript, we have extended our experiments to include comparisons with the impressive work SunGen [1], as presented in Table 1 in latest version. Indeed, SunGen, with its meta-learning-based approach, demonstrates greater robustness compared to Hu et al.'s method, making it a more suitable baseline for our study. However, our results show that both IMP-Loss and DIMP-Loss consistently outperform SunGen in overall performance. Additionally, we have included a computational time comparison in Appendix E, which highlights that SunGen's approach requires approximately seven times more computation time than our DIMP-Loss method. This reinforces the efficiency of our proposed approach.
>
> Furthermore, we have added UniGen [2] and FuseGen [3] in the latest version of the Related Work section, with updates highlighted in red for clarity. UniGen employs contrastive learning with the generator's confidence scores, which are not applicable in our setting due to our reliance on the closed-source LLM GPT-3.5; therefore, we cannot conduct the pseudo-relabelling mentioned in UniGen. FuseGen, on the other hand, generates synthetic data from multiple LLMs and uses a multi-stage regeneration process, culminating in a boosting technique to adjust data weights. This approach is interesting and inspiring; however, for single LLM, it seems unable to show its potentials on the classification task.  We did some pilot study, using their implementation on financial data generated from a single source (GPT-3.5), and revealed limitations in FuseGen's effectiveness in this setting. The optimal testing accuracy achieved was 67.99%, significantly lower than the 77.39% accuracy achieved with CE-Loss. Additionally, the computational cost of boosting is still considerable, as it requires inference over the entire training dataset during each epoch to compute errors for weight adjustment. In contrast, our method calculates weights once before training, drastically reducing computational overhead.
>
> In summary, we believe our approach offers a more efficient, robust, and versatile solution for tasks of this nature, with the extra advantage of compatibility with any loss function using common gradient optimization, such as SGD. Therefore, our approaches can also be used on UniGen or FuseGen, and potentially complement and enhance these frameworks.

---

> > ### Author Response · Authors · 2024-11-20
> > **Reply to Reviewer eiLk (Part 2/2)**
> >
> > ## Reference
> > [1] Gao et al., Self-Guided Noise-Free Data Generation for Efficient Zero-Shot Learning, ICLR 2023.
> >
> > [2] Choi et al., UniGen: Universal Domain Generalization for Sentiment Classification via Zero-Shot Dataset Generation, arXiv Preprint 2024.
> >
> > [3] Zou et al., FuseGen: PLM Fusion for Data-Generation-Based Zero-Shot Learning, arXiv Preprint 2024.

---

> > > ### Comment · Reviewer_eiLk · 2024-11-21
> > >
> > > Thank you for your patient rebuttal. I would like to note that I was satisfied with the revisions made by the authors, and I revised my review and rating accordingly.

---

### Official Review · Reviewer_R1bf · 2024-10-29

**Soundness:** 3
**Presentation:** 3
**Contribution:** 3
**Rating:** 8
**Confidence:** 3

**Summary:**

Since the development of LLMs, LLM-generated data are widely used for training models. However, the distribution of such artificial datasets may be misaligned from real-world datasets, creating concerns.
Therefore, during the training process, to prioritize data that is more diverse and has higher quality, the authors develop an approach to weighing LLM-generated data according to its quality and diversity. Compared to previous data-filtering techniques that only use a subset of generated data, this approach leverages all the training data while giving nonuniform weights to the loss of each data point.
Specifically, the authors theoretically define two losses: IMP-Loss and DIMP-Loss. IMP-Loss applies WCE-Loss, where the weight function is compounded by a quality checker (approximated with P, estimated by fitting a model on D_P, the real-world dataset) and a diversity checker (approximated with the inverse of Q, estimated by fitting a model on D_Q, which is the synthetic dataset). DIMP-Loss simplifies the calculation by estimating the diversity checker with the current model in training.
As a result, the performance using the newly devised loss improves compared to the baselines, with DIMP Loss falling slightly behind IMP loss, as a compromise for lower costs.

**Strengths:**

1. The theoretical derivation of IMP and DIMP loss is clear and sufficient.
2. The empirical improvement shown in table 1 is consistent.
3. There are multiple studies to verify the robustness of the method: different datasets in figure 1, different model sizes in table 2, and using various percentages of the training set in figure 2.
4. Section 3.2 shows the motivation for employing diversity and quality in the weighting strategy, laying the foundation for the methodology developed later in the section.
5. The writing and structure is clear to follow.

**Weaknesses:**

1. The results are derived using a BERT-Based model, which is quite small compared to contemporary LLMs. How would the method scale to larger LLMs? How would the costs grow as the model size grows?

**Questions:**

1. For the metrics "diversity" and "quality", would it be possible to verify that they actually correspond to what they refer to? I.e., does higher quality data actually get higher quality metrics? Currently, it seems like a term for the mathematical expression. While I understand it intuitively, it would also be nice to verify them empirically.
2. In the tweet irony case in Figure 1, DIMP loss start out with worse performance than other baselines for the first and second epoch. Could you explain why such a scenario would occur? Does this mean that these methods introduce instability to the training process?

---

> ### Author Response · Authors · 2024-11-20
> **Reply to Reviewer R1bf**
>
> We sincerely thank Reviewer R1bf for reviewing our submission and for recognizing the clarity of our theoretical derivations, the robustness of our results, and the structure of our paper. Below, we address your comments in detail.
> ## Reply to Q1
> > For the metrics "diversity" and "quality", would it be possible to verify that they actually correspond to what they refer to? I.e., does higher quality data actually get higher quality metrics? Currently, it seems like a term for the mathematical expression. While I understand it intuitively, it would also be nice to verify them empirically.
>
> For quality, we consider data points that are more relevant, and for diversity, we consider those data appear less frequently. In Appendix G.3 of new revision, depicts three scenarios characterized by lower quality (random labels, unrelated inputs) and reduced diversity (repeated data). Furthermore, Figure 4 presents the average scores of the Diversity Checker and Quality Checker, as well as the weight assigned to each of the three scenarios in comparison to real-world data. We appreciate your attention to these details and hope this additional clarification addresses any potential concerns.
>
> ## Reply to Q2
> > In the tweet irony case in Figure 1, DIMP loss start out with worse performance than other baselines for the first and second epoch. Could you explain why such a scenario would occur? Does this mean that these methods introduce instability to the training process?
>
> Thank you for your observation, in DIMP-Loss, we use a random initialized diversity checker, so the variance in the begining stages is expected. However, from the result of experiments, we can see the variance of DIMP-Loss converges in later epochs.
>
> ## Reply to Weakness 1
> > The results are derived using a BERT-Based model, which is quite small compared to contemporary LLMs. How would the method scale to larger LLMs? How would the costs grow as the model size grows?
>
> We appreciate your observation regarding the potential of our method to be applied to larger LLMs. In this work, our focus is on fine-tuning text classifiers on LLM-generated data. To explore scalability, we conducted an empirical study using BERT-Large as the fine-tuning model while employing a smaller model as the quality checker, as shown in Table 2. The results demonstrate that DIMP-Loss achieves significant improvements compared to CE-Loss, indicating the method's potential to scale effectively to larger models without requiring an equally large model for the quality checker. This makes our approach more practical and cost-efficient. We believe it would be exciting to further explore the application of our method in training larger models.

---

> > ### Comment · Reviewer_R1bf · 2024-11-25
> > **Thank you**
> >
> > The author's rebuttal has addressed my concerns. I will raise my score.

---

### Official Review · Reviewer_gCSy · 2024-11-10

**Soundness:** 3
**Presentation:** 3
**Contribution:** 2
**Rating:** 6
**Confidence:** 2

**Summary:**

This paper introduced IMP-Loss and DIMP-Loss as novel weighted-loss objectives to enhance the performance of models trained on LLM-generated data. Paper showed results demonstrating that both methods outperformed traditional loss functions CE-loss, focal loss, Meta-learning weighting on 3 NLP tasks Financial Phrasebank, Tweet Irony and MRPC.

Both IMP-Loss and DIMP-Loss include quality over diversity as weighting per training example. Quality weight is the predicted conditinoal probability from a BERT-base model trained on a small real world data of size 200-400 samples. Diversity for IMP-Loss is similarly trained model but on synthetic data. Diversity for DIMP-Loss is the current model's predicted conditional probability. IMP-Loss is based on importance sampling, DIMP-loss is motivated by online batch selection methods.

For experiments, paper compared training on synthetic data alone, noisy synthetic data (label flipped, duplicated data entry). In both case, IMP-Loss and DIMP-Loss work better than the quality checker and three other loss functions. Paper also applied the two weighted loss function to real data, where the results showed better than training on real data using the other loss functions too.

**Strengths:**

- The  IMP-Loss and DIMP-Loss  approaches are novel contributions. The approaches are clearly motivated. Derivation are clearly demonstrated step by step. Approximations are used to make the implementation efficient.
- The experiments covered several scenarios, including small real data, synthetic data, noisy data and real data. In all cases, the introduced methods work best, showing the robustness of the approaches. Several oblations are interesting: increasing real data for quality checker increases final performance. Diversity check is very important for final performance.

**Weaknesses:**

The paper introduces two weighting scheme for synthetic data. The experiments can be made more convincing:
- The three benchmarks can be expanded to more NLP tasks. Eg nn the ZEROGEN paper cited here there are several benchmarks: sQuAD, QNLI and AdversarialQA, which are particularly challenging for synthetic data generated from LLM. The performance was far below supervised results. It will be interesting to apply the weighting method to see if the gap can be reduced for these challenging benchmarks.
- The paper used 5 epochs for training the downstream models by Line 1101. It is unclear whether the models have converged. It is interesting to not only see weighting proposed here can speed up learning, but what happens at convergence.
- The weighting method does not seem to achieve SOTA for the two benchmarks. It would be more convincing if the paper adopted a strong baseline rather than BERT-large to build on top and compare with SOTA results.
  - mrpc https://paperswithcode.com/sota/semantic-textual-similarity-on-mrpc, tinybert is at 86.4%.
  - financial phrasebank https://paperswithcode.com/sota/text-classification-on-financial-phrasebank. 96.7% for distillbert.

**Questions:**

- Paper assumes Q(x) ~ P(x) on line 171, but in line 154-164, the synthetic data has high diversity and entropy. Wondering how to interpret the two together?
- Synthetic data includes the label for the data. For Q(x) ~ P(x) on line 171, does that means Q(x, y) ~ P(x, y)? Seems the paper mix Q(x, y), Q(y|x) eg at line 942 too.
- Line 999 missing ', so it should be x_{p'}, right?
- The table 1 shows Focal-loss and Hu et al.'s method did not work better than CE-Loss, esp for the "large real world" case, could you comment on the reason?
- For the DIMP-Loss at line 274, if we apply it to large real data, where P' is also trained on the same large real data. In this case, the weighting will converge to 1, is that correct? In that case, the final performance at convergence should be the same as CE-Loss, right? How does this case affect table 1?

---

> ### Author Response · Authors · 2024-11-20
> **Reply to Reviewer gCSy (Part 1/2)**
>
> We sincerely thank Reviewer gCSy for thoroughly reviewing our submission and appreciating the clarity, robustness, and structure of our work. Below, we provide detailed responses to your comments.
>
> ## Reply to Q1
> > Paper assumes Q(x) ~ P(x) on line 171, but in line 154-164, the synthetic data has high diversity and entropy. Wondering how to interpret the two together?
>
> Thank you for pointing out the potential confusion, the high conditional entropy in $Q(y|x)$ reflects the diversity of the synthetic data and may include label noise, as noted in Section 3.2. This does not contradict the assumption $P(x) \approx Q(x)$ since the assumption pertains solely to the marginal input distributions.
>
> Additionally, this assumption is required solely for IMP-Loss and does not impact the derivation of DIMP-Loss.
> ## Reply to Q2
> > Synthetic data includes the label for the data. For Q(x) ~ P(x) on line 171, does that means Q(x, y) ~ P(x, y)? Seems the paper mix Q(x, y), Q(y|x) eg at line 942 too. Line 999 missing ', so it should be x_{p'}, right?
>
> Thank you for pointing out the potential confusion in our notation. The assumption $Q(x) \approx. P(x)$ pertains solely to the **input distribution**, not the joint distribution. It is possible that $Q(x, y) = Q(x)Q(y|x) \neq P(x)P(y|x) = P(x, y)$ due to $P(y|x) \neq Q(y|x)$. In line 942, $Q(x, y)$ refers to the joint distribution, and the subsequent reference to $Q(x)$ isolates the input distribution since $Q(x, y) = Q(y|x)Q(x)$, and $Q(y|x)$ is eliminated in the denominator of the weight. The expectation remains over the joint distribution $Q(x, y)$. We will refine our notation in the next version to make this clearer, ensuring $Q$ and $P$ consistently denote joint distributions, and revising $H(P)$ to $H_P(Y|X)$ in Section 3.2 to explicitly indicate conditional entropy.
>
> Lastly, you are correct about the typographical issue in line 999. The term should indeed be $x_{p'}$, and we have already updated this in our edits. Thank you for your careful reading and valuable feedback.
> ## Reply to Q3
> > The table 1 shows Focal-loss and Hu et al.'s method did not work better than CE-Loss, esp for the "large real world" case, could you comment on the reason?
>
> Focal Loss prioritizes more diverse or difficult data points, which can be beneficial in certain out-of-distribution scenarios, such as handling imbalanced labels. However, it does not explicitly account for the **quality** of the data, which may result in inadvertently prioritizing noisy or mislabelled data points during training, reducing its effectiveness compared to CE-Loss in such cases.
>
> Additionally, in our opinion, Hu et al.'s method can be viewed as a bilevel optimization approach [1] that adjusts data weights while training the model. However, in practice, it often exhibits unstable behaviour and is highly sensitive to hyperparameters [2]. In our experiments, we used the code provided by the authors of the paper and tested several hyperparameter configurations, including the default settings. The best results from these experiments are reported in the table. To provide a more comprehensive comparison, we conducted additional experiments on another meta-learning-based method, SunGen [3], as shown in Table 1 in this revision. The results demonstrate that while SunGen is able to surpass CE-Loss in most cases, its performance remains slightly lower than that of IMP-Loss and DIMP-Loss. Furthermore, as shown in Table 3, SunGen requires approximately seven times the computational time compared to our methods.
>
> ## Reference
> [1] Can (Sam) Chen, et al., Gradient-based Bi-level Optimization for Deep Learning: A Survey, 2023
> [2] Minyoung Kim, et al., A Stochastic Approach to Bi-Level Optimization for Hyperparameter Optimization and Meta Learning, 2024
> [3] Gao et al., Self-Guided Noise-Free Data Generation for Efficient Zero-Shot Learning, ICLR 2023.

---

> ### Author Response · Authors · 2024-11-20
> **Reply to Reviewer gCSy (Part 2/2)**
>
> ## Reply to Question 4
> > For the DIMP-Loss at line 274, if we apply it to large real data, where P' is also trained on the same large real data. In this case, the weighting will converge to 1, is that correct? In that case, the final performance at convergence should be the same as CE-Loss, right? How does this case affect table 1?
>
> Indeed, in large real-world scenarios, if the training dataset and the dataset used for the quality checker are sufficiently large, the weights from both DIMP-Loss and IMP-Loss will converge to 1. In this case, the performance at convergence would align with CE-Loss. However, in our experiments, the training dataset size was around 3000 samples, which may have a distribution gap compared to the real-world distribution. In our view, the quality checker and diversity checker play crucial roles in mitigating overfitting on the training data. By focusing on minimizing real-world loss rather than just training loss.
>
>
> ## Reply to Weakness 1
> > The three benchmarks can be expanded to more NLP tasks. Eg nn the ZEROGEN paper cited here there are several benchmarks: sQuAD, QNLI and AdversarialQA, which are particularly challenging for synthetic data generated from LLM. The performance was far below supervised results. It will be interesting to apply the weighting method to see if the gap can be reduced for these challenging benchmarks.
>
> We appreciate you noticed our methods had the potentials to apply on other NLP tasks, but we need to collect more synthetic data from LLMs for these challenging benchmarks; therefore, we focused on text-classification task due to limited budget.
>
> ## Reply to Weakness 2
> > The paper used 5 epochs for training the downstream models by Line 1101. It is unclear whether the models have converged. It is interesting to not only see weighting proposed here can speed up learning, but what happens at convergence.
>
> Thank you for your insightful observation. In Appendix E, we tried to compare the computational time, so the numbers of epoch were fixed to 5. We also tested the performance in more epochs. For example, in our experimental setup of Table 1 (Appendix F), we initially tested 5 and 7 epochs, reporting the best accuracy run in Table 1.
>
> To address concerns about convergence, we conducted additional experiments with 10 epochs for the tweet benchmark and present the training dynamics. As shown in the plot from [anonymous link](https://i.ibb.co/fDYBQFL/training-dynamic-10-epoch.png), the results align with those in paper's Figure 1. Both IMP-Loss and DIMP-Loss exhibit low variations by the end of training, with testing accuracy converging to higher values than the baselines, even when the total numbers of epoch are doubled.
>
> Additionally:
> * It is worth noting that convergence occurs relatively late due to the use of a learning rate scheduler and warm-up.
> * In our experiments, we consistently used a small real-world dataset as a validation set to select the best checkpoint. This demonstrates that our method aligns well with standard model selection practices and remains practical for common scenarios.

---

### Meta-Review · Area_Chair_gyhy · 2024-12-16

**Metareview:**

This paper introduces IMP-Loss and DIMP-Loss, two novel weighted-loss functions designed to improve the performance of models trained on LLM-generated synthetic data. The approach addresses challenges such as the misalignment between real-world and synthetic data distributions and the limited availability of real-world data. There are a few key contributions. First, the Loss Function Design: author proposed IMP-Loss based on importance sampling, to assign weights using both quality and diversity metrics; authors also proposed DIMP-Loss, inspired by online batch selection methods, to simplify diversity estimation by using the current model’s predictions, trading slight performance for reduced computational cost. Both loss functions assign nonuniform weights to each data point instead of discarding data, unlike traditional filtering techniques. Quality and Diversity Metrics is another contribution. Derived from the conditional probabilities predicted by a BERT-base model trained on a small real-world dataset (200–400 samples), Quality Weights are obtained. Both losses are also showed to obtain Diversity Weights. Solid experimental Results are provided to show efficacy of proposed methods. Both IMP-Loss and DIMP-Loss consistently outperformed traditional loss functions (e.g., cross-entropy loss, focal loss, and meta-learning weighting) and even improved performance on real-world data. This study highlights how prioritizing quality and diversity in synthetic data training can bridge the performance gap between models trained on synthetic versus real-world data, making synthetic data more viable for machine learning tasks.


Strength of this paper

Novelty and Motivation: the introduction of IMP-Loss and DIMP-Loss. The motivations for incorporating quality and diversity in the weighting strategy are well-articulated and grounded in prior challenges with synthetic data.
Clarity and Structure: The paper is well-written and clearly structured, with a well-defined problem formulation and step-by-step derivation of the proposed methods.
Efficiency and Practicality: The use of approximations improves computational efficiency, making the approach more practical than meta-learning-based alternatives. Runtime complexity analysis demonstrates the applicability of the methods in real-world scenarios.
Empirical Validation: Comprehensive experiments validate the efficacy of the methods, with consistent performance improvements shown in results tables and figures. The empirical results are compelling, showing the superiority of the proposed methods over baselines across multiple scenarios.
Timeliness and Relevance: The work addresses a long-standing issue of noise in synthetic data, building on prior research while mitigating its limitations.

Weakness of this paper

Several reviewers raised few concerns/limitations of this paper. By addressing these limitations, the paper could strengthen its experiment and expand impact.
1. Convergence Uncertainty: The experiments use only five training epochs, and it is unclear whether the models have fully converged. Analyzing the proposed weighting method's impact on convergence speed and performance at full convergence would add depth to the results.
2. Scalability to Larger Models: The experiments rely on a BERT-base model, which is relatively small compared to contemporary large language models (LLMs). It is unclear how the method scales to larger models in terms of performance improvements and computational cost.
3. Novelty Concerns: The core idea of weighting synthetic data is not entirely new, as similar concepts have been explored (e.g., in "Self-Guided Noise-Free Data Generation for Efficient Zero-Shot Learning"). While the formulation and implementation here are distinct, the conceptual similarity somewhat limits the novelty.

**Additional Comments On Reviewer Discussion:**

Above summarized the strength and weaknesses raised by reviewers. Most of the weaknesses were addressed via further discussion and more experiment results. Given the relatively positive ratings, the strengthens summarized above, and mitigated concern on weaknesses, I recommend to accept this paper.

---

### Decision · Program_Chairs · 2025-01-22

Accept (Spotlight)